# From Human Attention to Diagnosis: Semantic Patch-Level Integration of Vision-Language Models in Medical Imaging

**Dmitry Lvov**
Research Center of
the Artificial Intelligence Institute
Innopolis University
Innopolis, Russia 420500
d.lvov@innopolis.ru

**Ilya Pershin**
Research Center of
the Artificial Intelligence Institute
Innopolis University
Innopolis, Russia 420500
i.pershin@innopolis.ru

## Abstract

Predicting human eye movements during goal-directed visual search is critical for enhancing interactive AI systems. In medical imaging, such prediction can support radiologists in interpreting complex data, such as chest X-rays. Many existing methods rely on generic vision–language models and saliency-based features, which can limit their ability to capture fine-grained clinical semantics and integrate domain knowledge effectively. We present **LogitGaze-Med**, a state-of-the-art multimodal transformer framework that unifies (1) domain-specific visual encoders (e.g., CheXNet), (2) textual embeddings of diagnostic labels, and (3) semantic priors extracted via the logit-lens from an instruction-tuned medical vision–language model (LLaVA-Med). By directly predicting continuous fixation coordinates and dwell durations, our model generates clinically meaningful scanpaths. Experiments on the GazeSearch dataset and synthetic scanpaths generated from MIMIC-CXR and validated by experts demonstrate that LogitGaze-Med improves scanpath similarity metrics by 20–30% over competitive baselines and yields over 5% gains in downstream pathology classification when incorporating predicted fixations as additional training data.

## 1 Introduction

Understanding and predicting human eye movements during visual search is a long-standing problem in both cognitive science and computer vision [1–4]. In medical imaging, modeling expert gaze—such as how radiologists examine chest X-rays—offers new avenues to support diagnostic decision-making, enhance training, and develop interactive AI systems.

However, existing gaze prediction methods often fail in real-world clinical settings, where visual targets are subtle, abstract, and highly variable depending on the diagnostic task [5, 6]. While most prior work has focused on free-viewing or saliency-based gaze prediction [7, 8], these approaches typically neglect the observer's goal. In contrast, we study goal-directed gaze modeling: predicting the spatio-temporal sequence of fixations (scanpaths) that occur during active visual search for specific diagnostic targets [9].

This task poses unique challenges in medical imaging. Diagnostic categories like "pleural effusion" may not correspond to well-defined objects or regions [10, 11]. Visual cues can be ambiguous or distributed across anatomical structures. Moreover, collecting large-scale gaze data across all possible pathologies is infeasible due to cost, data availability, and privacy concerns [12].

39th Conference on Neural Information Processing Systems (NeurIPS 2025).

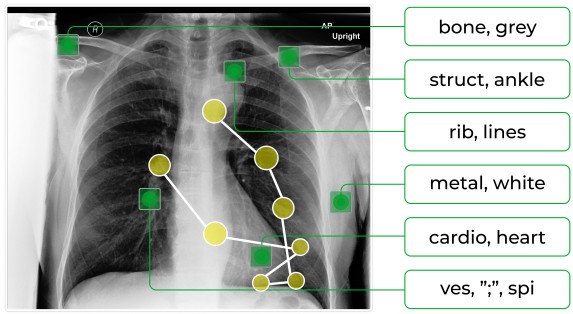

Figure 1: Scanpath visualization on a chest X-ray image labeled "normal", with fixations (yellow circles) indicating attention shifts. Image patches are semantically decoded via VLM into labels such as "bone, gray, metal", "cardio", and "heart", aligning gaze prediction with clinical visual understanding.

To address these challenges, we introduce **LogitGaze-Med**, a transformer-based [13] framework that integrates three complementary modalities: (1) visual features from a domain-specific encoder (e.g., CheXNet [14]); (2) text embeddings that represent diagnostic queries or labels; and (3) semantic patch-level priors extracted from a medical vision–language model (LLaVA-Med), using the logit lens technique [15–18]. As shown in Figure 1, these decoded keywords help align predicted scanpaths with clinically meaningful image regions.

We evaluate LogitGaze-Med on both the GazeSearch dataset [19] and a large-scale synthetic gaze dataset generated over MIMIC-CXR [20, 11], with synthetic scanpaths validated through expert human evaluation. Our model shows strong performance across standard scanpath similarity metrics and maintains robustness when tested with alternative medical VLMs (LLaVA-Rad [21]) and modern encoders (CheSS, PEAC [22, 23]). In addition, we find that our model's predicted fixations significantly improve downstream pathology classification, suggesting practical utility beyond gaze modeling. Our main contributions are:

- We introduce **LogitGaze-Med**, the first framework to apply logit-lens decoding to medical VLMs for clinically grounded patch-level gaze prediction, with comprehensive ablation studies demonstrating the importance of domain-specific components.

- Our formulation of scanpath prediction as continuous regression over spatial coordinates and dwell durations improves downstream pathology classification by over 5%.

- Expert human evaluation validates synthetic scanpaths with high realism (4.3/5.0) and clinical relevance scores (4.2/5.0).

## 2   Related Work

We review prior work in three relevant areas: scanpath prediction, VLMs(vision–language models), and interpretability.

**Scanpath Prediction**   Early models of scanpath generation relied on saliency maps or hand-crafted heuristics [24], lacking semantic understanding and goal-awareness. Recent transformer-based methods have advanced goal-directed gaze modeling. GazeFormer [9] introduced a zero-shot "ZeroGaze" task by encoding search goals via natural language, achieving strong spatial–temporal accuracy while being over five times faster than prior approaches. HAT(The Human Attention Transformer) [25] unified top-down and bottom-up attention in a single framework, leveraging a simplified foveated retina to model human-like spatio-temporal attention. GazeXplain [26] extended this by generating natural language rationales alongside fixation sequences, bridging "where" and "why" people look. LookHear (ART) [27] tackled multimodal gaze by modeling real-time fixations during spoken object reference. LogitGaze [28] integrated semantic priors from VLMs using logit-lens decoding, improving prediction accuracy by 15% and enhancing interpretability through explicit

concept-level alignment. Foundational insights such as inhibition of return (IOR) [29] continue to inform fixation dynamics by discouraging re-attending to previously viewed regions.

In the medical domain, GazeSearch [19] introduced the first task-aligned chest X-ray visual search dataset and a dedicated baseline ("ChestSearch"), revealing that general-purpose gaze models underperform in clinical settings. Despite these advances, few approaches jointly model clinical semantics, task conditioning, and probabilistic scanpaths in a unified, end-to-end architecture.

**Vision–Language Models**  Instruction-tuned VLMs have shown strong generalization across domains. LLaVA [15] bootstrapped multimodal instruction tuning using GPT-4 [30], demonstrating robust performance on open-ended vision tasks. LLaVA-Med [16] adapted this approach to biomedical images by generating self-instruction data from PubMed figure captions [31], outperforming prior Med-VQA models with minimal domain-specific fine-tuning. LLaVA-Rad [21] further specialized this approach for radiology, training on 697k image-report pairs. Voila-A [32] leveraged AR/VR gaze data to align VLM attention with human fixations, using GPT-4 to annotate the VOILA-COCO dataset and integrating gaze into perceiver modules for interpretability. R-LLaVA [33] injected region-level priors into CLIP inputs to enhance Med-VQA accuracy, emphasizing the utility of explicit visual context. These studies show the promise of task-aligned or gaze-aware VLMs, but most do not directly model spatio-temporal attention or fixations.

**Interpretability of VLMs**  Interpreting VLMs is essential for trustworthy AI, especially in medicine. Neo et al. [18] showed that LLaVA gradually refines object-level semantics across layers. The logit lens [17] projects intermediate activations into the output space, revealing how representations evolve—from generic concepts in early layers to clinical terms like "consolidation." Originally for language models, this method was adapted to vision-language settings in scan path prediction task [28], offering interpretable semantic priors.

## 3  Methodology

Our scanpath prediction pipeline consists of four main stages: (1) logit-lens semantic extraction; (2) preprocessing of visual and textual features; (3) joint transformer encoding with multiterm loss formulation; and (4) fixation decoding and scanpath regression. All components are aligned with the schematic shown in Figure 2.

### 3.1  Logit-Lens Semantic Extraction

Given an input chest X-ray $I \in \mathbb{R}^{H \times W}$, we extract patch-level semantics using a vision–language transformer. The image is divided into a $P \times P$ grid (with $P = 24$, yielding $M = 576$ patches), and a pretrained multimodal model—either LLaVA [15], LLaVA-Med [34], or LLaVA-Rad [21] fine-tuned on radiological data—produces the final hidden state for each patch:

$$h_i \in \mathbb{R}^d, \quad i = 1, \ldots, M.$$

After layer normalization, logits over the vocabulary are computed:

$$h_i' = \mathrm{LayerNorm}(h_i), \quad \ell_i = W_{\mathrm{lm}} h_i' + b_{\mathrm{lm}}, \quad p_i = \mathrm{softmax}(\ell_i) \in \mathbb{R}^V.$$

From the top-$k$ logits $\{j_{i,1}, \ldots, j_{i,k}\}$, we extract corresponding word vectors using the frozen language embedding matrix $E_{\mathrm{lm}} \in \mathbb{R}^{V \times d}$:

$$S_i = \left[ E_{\mathrm{lm}}(j_{i,1}), \ldots, E_{\mathrm{lm}}(j_{i,k}) \right] \in \mathbb{R}^{k \times d}.$$

These are pooled (e.g., via mean) into a single semantic vector $s_i \in \mathbb{R}^d$. This logit-lens mechanism [17, 18] exposes latent clinical concepts—such as "consolidation" or "effusion"—at the patch level.

### 3.2  Visual and Textual Feature Preprocessing

**Visual Feature Encoders.**  Each image is resized to $768 \times 768$, normalized, and passed through one of several CNN backbones:

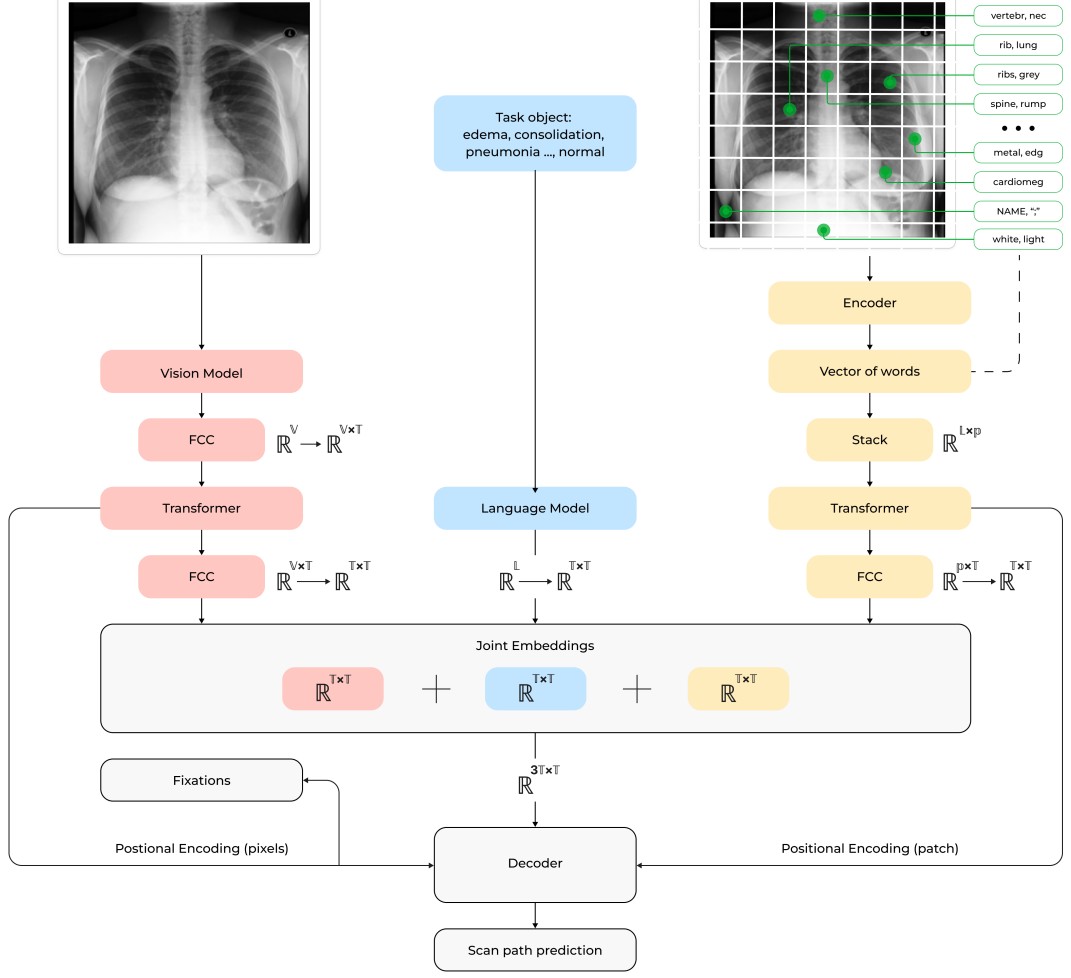

Figure 2: LogitGaze-Med architecture integrating vision-language processing with multiscale positional encoding. The framework fuses visual features (patch/pixel levels) with semantic embeddings (e.g., *bone, heart, opacity*) through a decoder. Joint optimization across classification tasks such as *edema*, *pneumonia*, and *normal*, along with spatial ($\mathcal{L}_x, \mathcal{L}_y$) and temporal ($\mathcal{L}_t$) losses, enables human-like scanpath generation. Closed-loop decoding projects hidden states into an interpretable clinical vocabulary space.

- **ResNetCOCO.** A ResNet50-FPN [35] from Mask R-CNN [36], pretrained on MS COCO [37], processes batches $\mathbf{x} \in \mathbb{R}^{B \times 3 \times 768 \times 768}$:

$$\mathbf{f} = \text{backbone}(\mathbf{x}) \in \mathbb{R}^{B \times C \times H' \times W'}, \quad C = 2048, \ H' \times W' = 24 \times 24.$$

  Flattening and permuting yields:

$$X_{\text{vis}} \in \mathbb{R}^{B \times M \times d_{\text{vis}}}, \quad d_{\text{vis}} = 2048.$$

- **CheXNet [14].** A DenseNet-121 [38], pretrained on the ChestX-ray14 dataset [39], outputs $\mathbb{R}^{B \times 1024 \times H' \times W'}$. A $1 \times 1$ convolution expands channels to 2048, yielding:

$$X_{\text{vis}} \in \mathbb{R}^{B \times M \times 2048}.$$

- **Alternative encoders.** We also evaluate modern medical image encoders including CheSS [22] and PEAC [23] to assess framework robustness across different visual backbones.

**Textual Task Embeddings.** Each diagnostic label (e.g., "normal," "pneumonia") is encoded with a SentenceTransformer (e.g., `stsb-roberta-base-v2`) [40]:

$$e_{\text{task}} \in \mathbb{R}^{d_{\text{text}}}, \quad t = W_t \, e_{\text{task}} + b_t \in \mathbb{R}^d.$$

### 3.3 Joint Transformer Encoding and Loss Formulation

**Embedding Fusion.** Visual features $U_{\text{vis}} \in \mathbb{R}^{B \times M \times d}$ and semantic embeddings $U_{\text{sem}} \in \mathbb{R}^{B \times M \times d}$ are linearly projected into a shared dimension $d$. These are concatenated:

$$F_{\text{joint}} = \left[\, U_{\text{vis}} \,;\, U_{\text{sem}} \,\right] \in \mathbb{R}^{B \times 2M \times d}.$$

Learnable 2D positional encodings and broadcasted task embedding $T_{\text{task}} \in \mathbb{R}^{B \times d}$ are added:

$$Z = \text{TransformerEncoder}(F_{\text{joint}} + \text{PE}_{\text{patch}} + T_{\text{task}} \oplus \cdots \oplus T_{\text{task}}) \in \mathbb{R}^{B \times 2M \times d}.$$

**Loss Functions.** We jointly optimize three objectives with fixed weighting coefficients $\lambda_1$ and $\lambda_2$:

$$\mathcal{L} = \underbrace{\mathcal{L}_{\text{cls}}}_{\substack{\text{token} \\ \text{classification}}} + \lambda_1 \underbrace{\mathcal{L}_{\text{spatial}}}_{\substack{\text{coordinate} \\ \text{regression}}} + \lambda_2 \underbrace{\mathcal{L}_{\text{time}}}_{\substack{\text{dwell time} \\ \text{regression}}}.$$

The weighting parameters are fixed throughout training, with baseline values $\lambda_1 = 1.0$ and $\lambda_2 = 1.0$ selected via preliminary validation.

- **Token Classification:** At each timestep $t \in [1, L]$, the decoder predicts a binary stop token $\hat{m}_t \in [0,1]$. A masked cross-entropy loss is used:

$$\mathcal{L}_{\text{cls}} = -\frac{1}{N_{\text{valid}}} \sum_{t=1}^{L} \mathbb{I}_t \, \log \hat{m}_t^{(c_t)}, \quad c_t \in \{0, 1\}.$$

- **Spatial Loss:** We compute the L1 distance between predicted and ground truth coordinates:

$$\mathcal{L}_{\text{spatial}} = \frac{1}{N_{\text{valid}}} \sum_{t=1}^{L} \mathbb{I}_t \left( |\hat{x}_t - x_t| + |\hat{y}_t - y_t| \right).$$

- **Temporal Loss:** The predicted dwell time $\hat{\tau}_t$ is supervised using mean squared error:

$$\mathcal{L}_{\text{time}} = \frac{1}{N_{\text{valid}}} \sum_{t=1}^{L} \mathbb{I}_t \left( \hat{\tau}_t - \tau_t \right)^2.$$

### 3.4 Fixation Decoding and Scanpath Regression

**Decoder Architecture.** We use a transformer decoder with $N_{\text{dec}}$ layers and $L$ learnable query embeddings $\{Q_0, \ldots, Q_{L-1}\} \subset \mathbb{R}^d$. The initial query $Q_0$ is modulated with the first human fixation via 2D positional encoding, ensuring all compared methods are equally conditioned on initial gaze position. Self- and cross-attention operations over encoder outputs $Z$ produce:

$$F_{\text{dec}} \in \mathbb{R}^{B \times L \times d}.$$

**Fixation Parameter Regression.** For each step $t$, six MLP heads predict the spatial means $\mu_{x_t}, \mu_{y_t}$, dwell time mean $\mu_{\tau_t}$, and log-variances $\lambda_{x_t}, \lambda_{y_t}, \lambda_{\tau_t}$. Predictions are sampled via the reparameterization trick:

$$\hat{x}_t = \mu_{x_t} + \epsilon_{x_t} \exp(0.5 \, \lambda_{x_t}), \quad \epsilon_{x_t} \sim \mathcal{N}(0, 1),$$

and similarly for $\hat{y}_t$ and $\hat{\tau}_t$. This stochastic sampling enables the model to represent variability in human gaze behavior.

## 4 Results

### 4.1 Database

We evaluate *LogitGaze-Med* on several datasets that capture radiologists' eye movements during chest X-ray interpretation. Preference was given to datasets aligned with diagnostic search tasks.

Our primary dataset is *GazeSearch* [19], designed for scanpath prediction in medical imaging. Unlike earlier datasets that reflect free viewing, GazeSearch employs a target-present search paradigm with

known findings (e.g., cardiomegaly). It filters fixations to emphasize task-relevant sequences, making scanpaths more representative of purposeful diagnostic behavior.

To assess generalization, we also use the Eye Gaze dataset [11], which contains expert fixations but is limited in scale. To overcome this, we generate synthetic scanpaths for a large MIMIC-CXR subset [20], enabling downstream tasks like gaze-informed pathology classification and large-scale evaluation.

For cross-domain benchmarking, we include *COCO-Search18* [41], a visual search dataset from the general vision domain. LogitGaze-Med performs worse here, validating its specialization for medical imaging and the need for task-specific inductive biases.

## 4.2 Evaluation Metrics

To evaluate scanpath prediction, we follow the GazeSearch protocol [19] and report multiple metrics that capture spatial, temporal, and dynamic gaze characteristics.

**ScanMatch** [42] aligns scanpaths using a variant of the Needleman-Wunsch algorithm [43]. Each fixation is encoded by spatial bin and temporal order, with optional inclusion of duration(w/ Dur.). For "w/o duration"(w/o Dur.) variants, the alignment omits fixation length information.

**SED** [44, 45] (Scanpath Edit Distance) is a Levenshtein-based metric that counts insertions, deletions, and substitutions between discretized fixations. While duration and geometry are ignored, it remains an intuitive measure of sequence similarity.

**STDE** [46] embeds scanpaths into a temporal-spatial space and is sensitive to rhythm and timing variations. It captures biologically plausible gaze behavior, common in diagnostic reading.

**MultiMatch** [47] compares scanpaths along five continuous dimensions: shape, direction, length, position, and duration.

For downstream classification tasks, we also report AUROC (Area Under the Receiver Operating Characteristic Curve) to assess the effectiveness of gaze-informed models in distinguishing between clinical conditions. AUROC serves as a proxy for clinical utility and interpretability.

## 4.3 Scanpath Prediction

We evaluate scanpath prediction on the GazeSearch dataset [19], focusing on the target present condition. Our model, **LogitGaze-Med**, is compared against GazeFormer [9], HAT [25], ChestSearch [19], and LogitGaze [28].

LogitGaze-Med leverages LLaVA-Med [34] and optionally integrates CheXNet [14] for pathology-aware encoding. All models are trained from scratch on the GazeSearch dataset and identically conditioned on the first fixation to ensure fair comparison. All models are trained for 100 epochs (batch size 32) on a 3090Ti GPU. We use a 6-layer encoder-decoder transformer (hidden size 512), a staged learning rate schedule ($1e-6$ / $2e-6$ / $1e-4$), and dropout (0.4 on the classifier).

LogitGaze-Med outperforms all baselines on both sequence-level and component-wise metrics. With CheXNet features, we observe a relative ScanMatch gain of +26% (w/o duration) and +48% (w/ duration), and SED is reduced by 4%. As shown in Tables 1 and 2, improvements in vector and duration similarity confirm enhanced alignment with human scan dynamics.

Table 1: Performance on scanpath similarity metrics (higher is better for ScanMatch/STDE, lower is better for SED).

| Method | ScanMatch ↑ | | SED ↓ | STDE ↑ |
| --- | --- | --- | --- | --- |
| | w/o Dur. | w/ Dur. | | |
| GazeFormer [9] | $0.293 \pm 0.021$ | $0.201 \pm 0.015$ | $5.11 \pm 0.08$ | $0.799 \pm 0.004$ |
| HAT [25] | $0.309 \pm 0.020$ | – | $5.07 \pm 0.07$ | $0.800 \pm 0.004$ |
| GazeSearch [19] | $0.332 \pm 0.019$ | $0.223 \pm 0.014$ | $4.88 \pm 0.06$ | $0.809 \pm 0.004$ |
| LogitGaze [28] | $0.328 \pm 0.018$ | $0.225 \pm 0.015$ | $5.07 \pm 0.07$ | $0.810 \pm 0.004$ |
| LogitGaze-Med (Res) | $0.416 \pm 0.017$ | $0.325 \pm 0.012$ | $4.68 \pm 0.05$ | $0.852 \pm 0.003$ |
| **LogitGaze-Med (CheX)** | $\mathbf{0.419 \pm 0.016}$ | $\mathbf{0.330 \pm 0.010}$ | $\mathbf{4.68 \pm 0.05}$ | $\mathbf{0.855 \pm 0.003}$ |

Table 2: MultiMatch similarity (higher is better) across five components.

| Method | Vector | Direction | Length | Position | Duration |
|---|---|---|---|---|---|
| GazeFormer [9] | $0.902 \pm 0.008$ | $0.644 \pm 0.010$ | $0.899 \pm 0.009$ | $0.803 \pm 0.007$ | $0.595 \pm 0.015$ |
| HAT [25] | $0.909 \pm 0.007$ | $0.649 \pm 0.010$ | $0.910 \pm 0.008$ | $0.825 \pm 0.006$ | – |
| GazeSearch [19] | $0.917 \pm 0.006$ | $\mathbf{0.679 \pm 0.010}$ | $0.917 \pm 0.007$ | $0.829 \pm 0.006$ | $0.695 \pm 0.014$ |
| LogitGaze [28] | $0.882 \pm 0.009$ | $0.643 \pm 0.008$ | $0.923 \pm 0.005$ | $0.809 \pm 0.006$ | $0.625 \pm 0.013$ |
| LogitGaze-Med (Res) | $0.935 \pm 0.004$ | $0.650 \pm 0.008$ | $0.939 \pm 0.006$ | $0.823 \pm 0.005$ | $0.743 \pm 0.010$ |
| **LogitGaze-Med (CheX)** | $\mathbf{0.938 \pm 0.004}$ | $0.651 \pm 0.009$ | $\mathbf{0.948 \pm 0.005}$ | $\mathbf{0.823 \pm 0.005}$ | $\mathbf{0.740 \pm 0.010}$ |

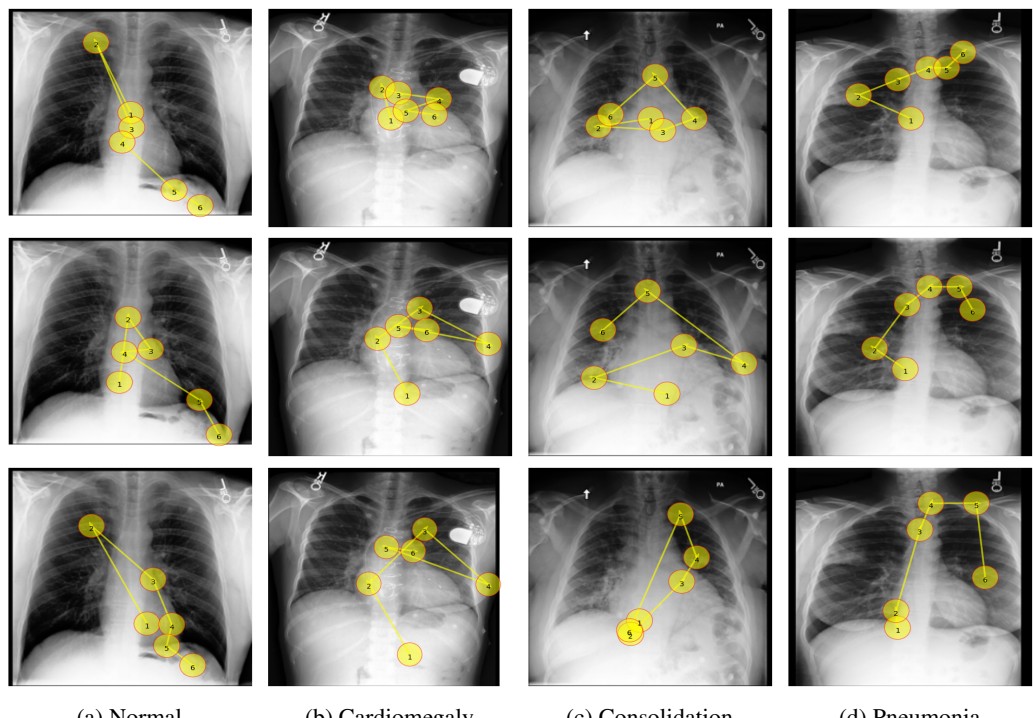

(a) Normal     (b) Cardiomegaly     (c) Consolidation     (d) Pneumonia

Figure 3: Comparison of human scanpaths (top), LogitGaze-Med predictions (middle), and Gaze-Search predictions (bottom) across tasks: *normal*, *cardiomegaly*, *consolidation*, and *pneumonia*. The models are evaluated under target-present conditions.

Figure 3 presents qualitative comparisons of scanpaths for several diagnostic tasks. Each triplet shows the human scanpath (top), predictions from LogitGaze-Med (middle), and predictions from Gaze-Search (bottom). Visually, LogitGaze-Med produces more human-like trajectories, with smoother saccades and task-relevant fixations that better reflect real radiological reading behavior. In contrast to baselines, it captures the diagnostic flow more accurately. These examples highlight the model's ability to generate plausible and clinically coherent visual search patterns.

## 4.4 X-Ray classification analysis

To assess the utility of gaze information in medical imaging, we reproduced and extended the classification setups introduced by [11] on the MIMIC-CXR dataset [20]. In addition to their original settings, we introduced a new variant using model-generated fixations, allowing us to evaluate generalization beyond human gaze annotations. We addressed a 3-class classification task (normal, congestive heart failure (CHF), and pneumonia) and explored how different gaze-informed architectures affect performance.

The baseline model uses a convolutional encoder and a linear classification head, taking only X-ray images as input. In the second setup, we augment the image with temporal fixation heatmaps: the image is encoded into a visual vector $\mathbf{v}_{CXR}$, while each of $m$ gaze heatmaps passes through a separate

CNN. These are aggregated via a BiLSTM with self-attention into a gaze vector $\mathbf{u}_{gaze}$, and the final prediction is based on $[\mathbf{v}_{CXR}; \mathbf{u}_{gaze}]$. The third setup introduces a multi-task U-Net [48] that jointly predicts pathology and reconstructs a static gaze heatmap. An EfficientNet-B0 [49] encoder is used, with a classification head at the bottleneck and a decoder predicting the gaze map. The total loss is a weighted sum of binary cross-entropy for both tasks.

To test generalization, we replaced human fixations with synthetic ones generated by various models (GazeFormer [9], LogitGaze [28], LogitGaze-Med), producing 8 scanpaths per image. Heatmaps derived from these were substituted into each pipeline without changing architecture or training protocol.

To validate the quality of our synthetic MIMIC-CXR scanpaths, we conducted a structured human evaluation with a board-certified radiologist (5+ years experience). The expert reviewed a hold-out set of 100 chest X-rays (50 with real and 50 with synthetic scanpaths, randomized and blinded) and provided ratings on a 5-point Likert scale for visual realism and clinical relevance.

Table 3: AUROC scores across three classification setups on MIMIC-CXR-JPG using human and synthetic gaze. For LogitGaze-Med, per-class AUROC scores are as follows: Baseline (Normal 0.87, CHF 0.85, Pneumonia 0.74), Temporal (Normal 0.93, CHF 0.92, Pneumonia 0.85), and U-Net (Normal 0.94, CHF 0.92, Pneumonia 0.87).

| Method | Baseline | Temporal | U-Net |
|---|---|---|---|
| Eye-Gaze [11] | 0.77 ± 0.02 | 0.82 ± 0.03 | 0.87 ± 0.02 |
| GazeFormer [9] | 0.78 ± 0.02 | 0.84 ± 0.02 | 0.89 ± 0.01 |
| LogitGaze [28] | 0.80 ± 0.01 | 0.87 ± 0.02 | 0.90 ± 0.01 |
| **LogitGaze-Med** | **0.82 ± 0.01** | **0.90 ± 0.02** | **0.91 ± 0.01** |

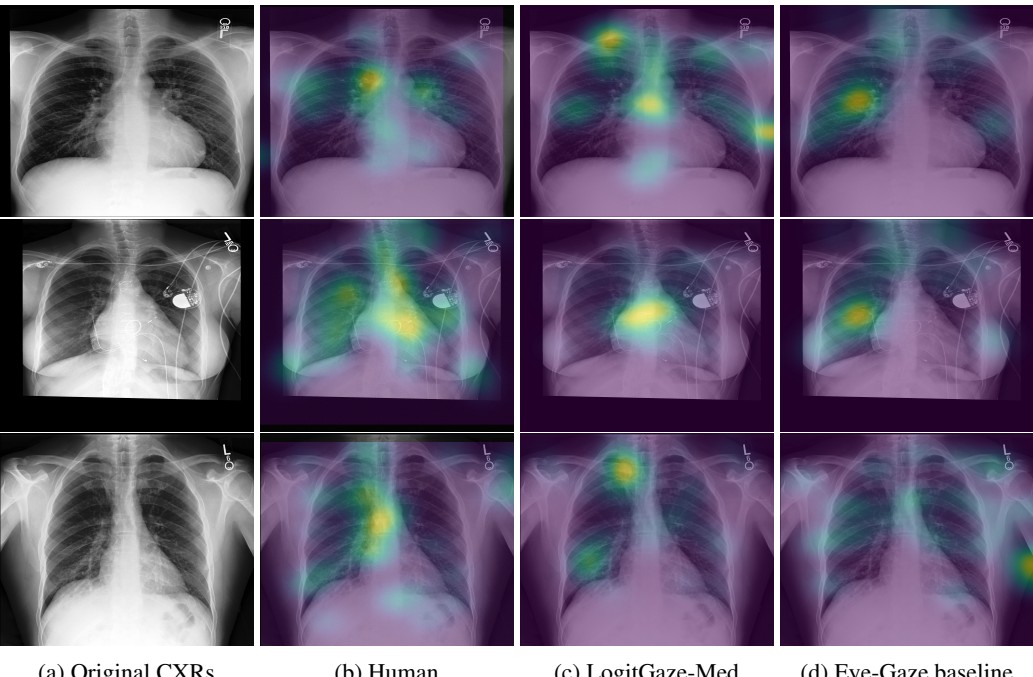

(a) Original CXRs     (b) Human     (c) LogitGaze-Med     (d) Eye-Gaze baseline

Figure 4: Attention maps for three conditions: normal (top), CHF (middle), and pneumonia (bottom). (a) Original X-rays; (b) Human gaze reference; (c) LogitGaze-Med; (d) Eye-Gaze baseline. Heatmaps generated by aggregating raw fixation coordinates into static heatmaps via Gaussian kernel smoothing [50]. Yellow denotes high attention.

The synthetic scanpaths received mean scores of 4.3±0.5 for visual realism and 4.2±0.6 for clinical relevance. In a binary classification task (real vs. synthetic), the expert achieved only 58% accuracy,

indicating that synthetic scanpaths were often indistinguishable from real ones. These results confirm the plausibility and clinical validity of our generated scanpaths.

Table 3 reports AUROC scores across all three setups. Temporal and multi-task models consistently outperform the baseline, showing that gaze information—especially when structured temporally—enhances diagnostic accuracy. Among gaze sources, LogitGaze-Med achieves the best performance in all cases, suggesting that its vision-language grounding improves semantic alignment with pathology.

Figure 4 contrasts attention maps from LogitGaze-Med and Eye-Gaze. LogitGaze-Med, limited to six predicted fixations as in GazeSearch [19], produces focused and pathology-aligned maps. In contrast, human and Eye-Gaze maps are often broader and less selective, reflecting more chaotic viewing behavior and weaker diagnostic relevance.

## 4.5 Ablation Analysis

We conduct comprehensive ablation experiments to evaluate the roles of different components, semantic continuity, and domain specificity in gaze prediction.

**Hyperparameter Sensitivity Analysis.** We analyzed the sensitivity of our model to the loss weighting parameters $\lambda_1$ (coordinate loss) and $\lambda_2$ (dwell time loss). As shown in Table 4, the model maintains robust performance across a broad range of values ($\lambda_1 \in [0.5 - 5.0]$, $\lambda_2 \in [0.1 - 5.0]$), with optimal performance at $\lambda_1 = 2.0$, $\lambda_2 = 0.5$. This configuration improves shape and position consistency while maintaining strong duration modeling.

Table 4: Effect of varying $\lambda_1$ (coordinate loss) and $\lambda_2$ (dwell time loss) on scanpath similarity (MultiMatch components). Best values in bold.

| Loss Weights | Shape | Direction | Length | Position | Duration |
|---|---|---|---|---|---|
| $\lambda_1 = 1.0, \lambda_2 = 1.0$ | 0.921 | 0.633 | 0.932 | 0.812 | 0.712 |
| $\lambda_1 = 2.0, \lambda_2 = 0.5$ | **0.938** | **0.651** | **0.948** | **0.823** | **0.740** |
| $\lambda_1 = 0.1, \lambda_2 = 1.0$ | 0.866 | 0.570 | 0.882 | 0.743 | 0.707 |
| $\lambda_1 = 1.0, \lambda_2 = 0.1$ | 0.917 | 0.624 | 0.926 | 0.805 | 0.610 |
| $\lambda_1 = 1.0, \lambda_2 = 5.0$ | 0.910 | 0.615 | 0.920 | 0.792 | 0.689 |
| $\lambda_1 = 5.0, \lambda_2 = 1.0$ | 0.887 | 0.593 | 0.902 | 0.760 | 0.701 |

**Component-wise Ablation.** Table 5 shows the importance of each modality. The image-only baseline performs poorly, confirming that visual features alone are insufficient. Adding clinical text substantially improves performance, highlighting the value of semantic intent. The full model with logit-lens achieves the best results, demonstrating the benefit of combining clinical context with localized attention.

Table 5: Stepwise ablation analysis of LogitGaze-Med on GazeSearch.

| Method | ScanMatch w/o Dur. ↑ | ScanMatch w/ Dur. ↑ | SED ↓ | STDE ↑ |
|---|---|---|---|---|
| Image only | $0.148 \pm 0.039$ | $0.126 \pm 0.034$ | $8.54 \pm 0.28$ | $0.562 \pm 0.094$ |
| + Text only | $0.280 \pm 0.020$ | $0.206 \pm 0.013$ | $5.95 \pm 0.06$ | $0.810 \pm 0.004$ |
| **LogitGaze-Med (full)** | **$0.419 \pm 0.016$** | **$0.330 \pm 0.010$** | **$4.68 \pm 0.05$** | **$0.855 \pm 0.003$** |

**Encoder and VLM Comparison.** Table 6 evaluates different combinations of VLMs and visual encoders. LLaVA-Med and LLaVA-Rad yield comparable results, confirming robustness to VLM choice. While modern encoders like CheSS and PEAC offer slight improvements, CheXNet remains competitive, supporting its use as a strong baseline.

**Semantic Continuity.** To assess the importance of structured semantic alignment, we compare LogitGaze-Med with two baselines: (1) a random predictor [27] and (2) a variant with shuffled reference alignments ($LogitGaze - Med_{shuf}$), which disrupts the scanpath's semantic structure. Table 7 shows that $LogitGaze - Med_{shuf}$ suffers a >40% drop in ScanMatch and a similar decline in MultiMatch, confirming that preserving alignment is essential for meaningful predictions.

Table 6: Performance comparison across different VLM and encoder combinations on GazeSearch dataset.

| Method | ScanMatch w/o Dur. ↑ | ScanMatch w/ Dur. ↑ | SED ↓ | STDE ↑ |
|---|---|---|---|---|
| LogitGaze-Med (ResNet) | $0.416 \pm 0.017$ | $0.325 \pm 0.012$ | $4.68 \pm 0.05$ | $0.852 \pm 0.003$ |
| LogitGaze-Med (CheXNet) | $0.419 \pm 0.016$ | $\mathbf{0.330 \pm 0.010}$ | $4.68 \pm 0.05$ | $0.855 \pm 0.003$ |
| LogitGaze-Med (CheSS) | $0.425 \pm 0.017$ | $0.321 \pm 0.012$ | $4.66 \pm 0.05$ | $0.857 \pm 0.003$ |
| LogitGaze-Med (PEAC) | $0.428 \pm 0.018$ | $0.319 \pm 0.012$ | $4.64 \pm 0.05$ | $0.858 \pm 0.004$ |
| LLaVA-Rad + CheXNet | $0.417 \pm 0.016$ | $0.329 \pm 0.010$ | $4.67 \pm 0.05$ | $0.854 \pm 0.004$ |
| LLaVA-Rad + CheSS | $0.426 \pm 0.018$ | $0.320 \pm 0.012$ | $4.65 \pm 0.05$ | $0.857 \pm 0.004$ |
| **LLaVA-Rad + PEAC** | $\mathbf{0.429 \pm 0.019}$ | $0.318 \pm 0.012$ | $\mathbf{4.63 \pm 0.05}$ | $\mathbf{0.859 \pm 0.004}$ |

Table 7: Effect of semantic continuity on gaze prediction.

| Method | ScanMatch w/o Dur | ScanMatch w/ Dur | MultiMatch Avg ↑ |
|---|---|---|---|
| LogitGaze-Med (CheX) | $\mathbf{0.419 \pm 0.016}$ | $\mathbf{0.330 \pm 0.010}$ | $\mathbf{0.820 \pm 0.009}$ |
| LogitGaze [28] | $0.328 \pm 0.017$ | $0.225 \pm 0.020$ | $0.776 \pm 0.011$ |
| Random [27] | $0.159 \pm 0.014$ | $0.148 \pm 0.016$ | $0.269 \pm 0.012$ |
| $LogitGaze\text{-}Med_{\text{shuf}}$ | $0.178 \pm 0.016$ | $0.158 \pm 0.015$ | $0.295 \pm 0.013$ |

**Domain Specificity.** We further evaluate the models on the COCO18 dataset [41] to analyze cross-domain generalization. While LogitGaze and GazeFormer were trained on COCO18, LogitGaze-Med was trained only on GazeSearch [19]. As shown in Table 8, LogitGaze-Med performs worse on COCO, reflecting its specialization for medical imagery. This gap highlights the importance of domain-aligned training for accurate scanpath prediction.

Table 8: Cross-domain evaluation on COCO18 dataset.

| Method | ScanMatch w/o Dur | ScanMatch w/ Dur | MultiMatch Avg ↑ |
|---|---|---|---|
| LogitGaze [28] | $\mathbf{0.527 \pm 0.012}$ | $\mathbf{0.454 \pm 0.013}$ | $\mathbf{0.862 \pm 0.008}$ |
| GazeFormer [9] | $0.492 \pm 0.014$ | $0.441 \pm 0.015$ | $0.816 \pm 0.010$ |
| LogitGaze-Med | $0.353 \pm 0.018$ | $0.289 \pm 0.020$ | $0.749 \pm 0.014$ |

## 5 Conclusion

We introduced **LogitGaze-Med**, a multimodal transformer that integrates domain-specific visual encoders, textual features, and semantic priors via logit-lens decoding from a medical VLM. Evaluated on real (GazeSearch) and validated synthetic (MIMIC-CXR) scanpaths, our model outperforms prior approaches, achieving 20–30% gains in scanpath similarity and over 5% improvement in downstream pathology classification.

Comprehensive evaluation demonstrates robustness across different medical VLMs and visual encoders. Expert human evaluation confirms the clinical plausibility of synthetic scanpaths with high realism (4.3/5.0) and clinical relevance scores (4.2/5.0).

Inference-time analysis shows a modest overhead (85 ms vs. 70 ms per sample) due to logit-lens extraction, maintaining suitability for clinical workflows with superior interpretability.

Key limitations include sensitivity to noisy activations from non-content tokens, as recent work[51] shows transformers encode important context in punctuation and function words, leading to potential fixation errors from unstable noun activations. Training on synthetic scanpaths also risks overfitting. Deployment in diverse clinical settings requires further validation across institutions and expertise levels.

We aim to deploy our framework in radiology for lesion tracking and training, with extensions to other medical imaging domains. Code will be released to support vision–language grounding and human–AI collaboration.

## Acknowledgments and Disclosure of Funding

This work was supported by the Ministry of Economic Development of the Russian Federation (agreement No. 139-10-2025-034 dd. 19.06.2025, IGK 000000C313925P4D0002).

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

# NeurIPS Paper Checklist

1. **Claims**

   Question: Do the main claims made in the abstract and introduction accurately reflect the paper's contributions and scope?

   Answer: [Yes]

   Justification: The abstract and introduction accurately reflect the main contributions of the paper: (1) the introduction of LogitGaze-Med, a domain-specific gaze prediction model leveraging semantic priors from a medical vision–language model; (2) consistent improvements in scanpath similarity metrics and downstream classification tasks; and (3) interpretability and domain specialization as key strengths. These claims are directly supported by experimental results and ablation analyses in Section 4.

   Guidelines:

   - The answer NA means that the abstract and introduction do not include the claims made in the paper.
   - The abstract and/or introduction should clearly state the claims made, including the contributions made in the paper and important assumptions and limitations. A No or NA answer to this question will not be perceived well by the reviewers.
   - The claims made should match theoretical and experimental results, and reflect how much the results can be expected to generalize to other settings.
   - It is fine to include aspirational goals as motivation as long as it is clear that these goals are not attained by the paper.

2. **Limitations**

   Question: Does the paper discuss the limitations of the work performed by the authors?

   Answer: [Yes]

   Justification: The limitations of the proposed method are discussed in the Conclusion section. In particular, we note that the logit-lens approach is sensitive to noisy or non-content activations, and that reliance on unstable content-word embeddings (e.g., from volatile tokens like nouns) can lead to fixation prediction errors in cluttered medical images. We also outline future directions to address these issues by incorporating syntactic signals such as punctuation and structure-related tokens.

   Guidelines:

   - The answer NA means that the paper has no limitation while the answer No means that the paper has limitations, but those are not discussed in the paper.
   - The authors are encouraged to create a separate "Limitations" section in their paper.
   - The paper should point out any strong assumptions and how robust the results are to violations of these assumptions (e.g., independence assumptions, noiseless settings, model well-specification, asymptotic approximations only holding locally). The authors should reflect on how these assumptions might be violated in practice and what the implications would be.
   - The authors should reflect on the scope of the claims made, e.g., if the approach was only tested on a few datasets or with a few runs. In general, empirical results often depend on implicit assumptions, which should be articulated.
   - The authors should reflect on the factors that influence the performance of the approach. For example, a facial recognition algorithm may perform poorly when image resolution is low or images are taken in low lighting. Or a speech-to-text system might not be used reliably to provide closed captions for online lectures because it fails to handle technical jargon.
   - The authors should discuss the computational efficiency of the proposed algorithms and how they scale with dataset size.
   - If applicable, the authors should discuss possible limitations of their approach to address problems of privacy and fairness.

- While the authors might fear that complete honesty about limitations might be used by reviewers as grounds for rejection, a worse outcome might be that reviewers discover limitations that aren't acknowledged in the paper. The authors should use their best judgment and recognize that individual actions in favor of transparency play an important role in developing norms that preserve the integrity of the community. Reviewers will be specifically instructed to not penalize honesty concerning limitations.

3. **Theory assumptions and proofs**

   Question: For each theoretical result, does the paper provide the full set of assumptions and a complete (and correct) proof?

   Answer: [NA]

   Justification: The paper does not contain formal theoretical results, theorems, or proofs. It focuses primarily on empirical evaluation and the application of a multimodal transformer-based framework for gaze prediction in medical diagnostics.

   Guidelines:

   - The answer NA means that the paper does not include theoretical results.
   - All the theorems, formulas, and proofs in the paper should be numbered and cross-referenced.
   - All assumptions should be clearly stated or referenced in the statement of any theorems.
   - The proofs can either appear in the main paper or the supplemental material, but if they appear in the supplemental material, the authors are encouraged to provide a short proof sketch to provide intuition.
   - Inversely, any informal proof provided in the core of the paper should be complemented by formal proofs provided in appendix or supplemental material.
   - Theorems and Lemmas that the proof relies upon should be properly referenced.

4. **Experimental result reproducibility**

   Question: Does the paper fully disclose all the information needed to reproduce the main experimental results of the paper to the extent that it affects the main claims and/or conclusions of the paper (regardless of whether the code and data are provided or not)?

   Answer: [Yes]

   Justification: We describe the architecture of LogitGaze-Med in detail (Section 3), including input representations, fusion strategies, and optimization setup. All datasets, evaluation protocols, and baseline implementations are fully specified (Section 4). We also provide instructions for processing both real and synthetic scanpaths.

   Guidelines:

   - The answer NA means that the paper does not include experiments.
   - If the paper includes experiments, a No answer to this question will not be perceived well by the reviewers: Making the paper reproducible is important, regardless of whether the code and data are provided or not.
   - If the contribution is a dataset and/or model, the authors should describe the steps taken to make their results reproducible or verifiable.
   - Depending on the contribution, reproducibility can be accomplished in various ways. For example, if the contribution is a novel architecture, describing the architecture fully might suffice, or if the contribution is a specific model and empirical evaluation, it may be necessary to either make it possible for others to replicate the model with the same dataset, or provide access to the model. In general. releasing code and data is often one good way to accomplish this, but reproducibility can also be provided via detailed instructions for how to replicate the results, access to a hosted model (e.g., in the case of a large language model), releasing of a model checkpoint, or other means that are appropriate to the research performed.
   - While NeurIPS does not require releasing code, the conference does require all submissions to provide some reasonable avenue for reproducibility, which may depend on the nature of the contribution. For example
     (a) If the contribution is primarily a new algorithm, the paper should make it clear how to reproduce that algorithm.

(b) If the contribution is primarily a new model architecture, the paper should describe the architecture clearly and fully.

(c) If the contribution is a new model (e.g., a large language model), then there should either be a way to access this model for reproducing the results or a way to reproduce the model (e.g., with an open-source dataset or instructions for how to construct the dataset).

(d) We recognize that reproducibility may be tricky in some cases, in which case authors are welcome to describe the particular way they provide for reproducibility. In the case of closed-source models, it may be that access to the model is limited in some way (e.g., to registered users), but it should be possible for other researchers to have some path to reproducing or verifying the results.

5. **Open access to data and code**

Question: Does the paper provide open access to the data and code, with sufficient instructions to faithfully reproduce the main experimental results, as described in supplemental material?

Answer: [Yes]

Justification: We commit to releasing our training and evaluation code, including detailed instructions and environment setup. All real scanpath data (GazeSearch) is already publicly available. For MIMIC-CXR, we describe how to obtain access and generate synthetic scanpaths using our pretrained model. All relevant scripts for preprocessing, training, and evaluation will be released alongside the paper.

Guidelines:

- The answer NA means that paper does not include experiments requiring code.
- Please see the NeurIPS code and data submission guidelines (`https://nips.cc/public/guides/CodeSubmissionPolicy`) for more details.
- While we encourage the release of code and data, we understand that this might not be possible, so "No" is an acceptable answer. Papers cannot be rejected simply for not including code, unless this is central to the contribution (e.g., for a new open-source benchmark).
- The instructions should contain the exact command and environment needed to run to reproduce the results. See the NeurIPS code and data submission guidelines (`https://nips.cc/public/guides/CodeSubmissionPolicy`) for more details.
- The authors should provide instructions on data access and preparation, including how to access the raw data, preprocessed data, intermediate data, and generated data, etc.
- The authors should provide scripts to reproduce all experimental results for the new proposed method and baselines. If only a subset of experiments are reproducible, they should state which ones are omitted from the script and why.
- At submission time, to preserve anonymity, the authors should release anonymized versions (if applicable).
- Providing as much information as possible in supplemental material (appended to the paper) is recommended, but including URLs to data and code is permitted.

6. **Experimental setting/details**

Question: Does the paper specify all the training and test details (e.g., data splits, hyperparameters, how they were chosen, type of optimizer, etc.) necessary to understand the results?

Answer: [Yes]

Justification: Section 4 (Results) provides a detailed description of datasets (GazeSearch and MIMIC-CXR), evaluation metrics, and experimental design. Hyperparameters, optimizer settings, learning rates, and data are described in Results section.

Guidelines:

- The answer NA means that the paper does not include experiments.
- The experimental setting should be presented in the core of the paper to a level of detail that is necessary to appreciate the results and make sense of them.

- The full details can be provided either with the code, in appendix, or as supplemental material.

7. **Experiment statistical significance**

   Question: Does the paper report error bars suitably and correctly defined or other appropriate information about the statistical significance of the experiments?

   Answer: [Yes]

   Justification: We report standard deviations for all evaluation metrics across experiments.

   Guidelines:
   - The answer NA means that the paper does not include experiments.
   - The authors should answer "Yes" if the results are accompanied by error bars, confidence intervals, or statistical significance tests, at least for the experiments that support the main claims of the paper.
   - The factors of variability that the error bars are capturing should be clearly stated (for example, train/test split, initialization, random drawing of some parameter, or overall run with given experimental conditions).
   - The method for calculating the error bars should be explained (closed form formula, call to a library function, bootstrap, etc.)
   - The assumptions made should be given (e.g., Normally distributed errors).
   - It should be clear whether the error bar is the standard deviation or the standard error of the mean.
   - It is OK to report 1-sigma error bars, but one should state it. The authors should preferably report a 2-sigma error bar than state that they have a 96% CI, if the hypothesis of Normality of errors is not verified.
   - For asymmetric distributions, the authors should be careful not to show in tables or figures symmetric error bars that would yield results that are out of range (e.g. negative error rates).
   - If error bars are reported in tables or plots, The authors should explain in the text how they were calculated and reference the corresponding figures or tables in the text.

8. **Experiments compute resources**

   Question: For each experiment, does the paper provide sufficient information on the computer resources (type of compute workers, memory, time of execution) needed to reproduce the experiments?

   Answer: [Yes]

   Justification: All experiments were conducted on a single NVIDIA RTX 3090 Ti GPU with 24 GB of VRAM. The computational requirements are moderate and reproducible using standard hardware.

   Guidelines:
   - The answer NA means that the paper does not include experiments.
   - The paper should indicate the type of compute workers CPU or GPU, internal cluster, or cloud provider, including relevant memory and storage.
   - The paper should provide the amount of compute required for each of the individual experimental runs as well as estimate the total compute.
   - The paper should disclose whether the full research project required more compute than the experiments reported in the paper (e.g., preliminary or failed experiments that didn't make it into the paper).

9. **Code of ethics**

   Question: Does the research conducted in the paper conform, in every respect, with the NeurIPS Code of Ethics `https://neurips.cc/public/EthicsGuidelines`?

   Answer: [Yes]

   Justification: We have carefully reviewed the NeurIPS Code of Ethics. Our research complies with all guidelines, including transparency, reproducibility, and the responsible use of data and models. All datasets used are publicly available and anonymized, and we disclose limitations and potential biases as appropriate.

Guidelines:

- The answer NA means that the authors have not reviewed the NeurIPS Code of Ethics.
- If the authors answer No, they should explain the special circumstances that require a deviation from the Code of Ethics.
- The authors should make sure to preserve anonymity (e.g., if there is a special consideration due to laws or regulations in their jurisdiction).

10. **Broader impacts**

Question: Does the paper discuss both potential positive societal impacts and negative societal impacts of the work performed?

Answer: [Yes]

Justification: Our work primarily focuses on improving the prediction of gaze patterns for medical image analysis, which can positively impact fields like radiology and assistive technologies. By improving gaze prediction, we may help doctors to better understand attention allocation when diagnosing medical images, potentially leading to more accurate diagnoses.

However, there are potential risks associated with this technology, particularly in sensitive domains such as medical diagnostics. Misuse of gaze prediction models could lead to biased decision-making if applied without proper safeguards. For instance, inaccurate gaze prediction could exacerbate existing biases in medical diagnostics if not rigorously tested across diverse patient populations.

To mitigate these risks, we advocate for transparent and responsible deployment of the technology, including continuous monitoring and validation of model outputs in real-world clinical settings. Additionally, we encourage collaboration with domain experts to ensure fairness and avoid unintended harms.

Guidelines:

- The answer NA means that there is no societal impact of the work performed.
- If the authors answer NA or No, they should explain why their work has no societal impact or why the paper does not address societal impact.
- Examples of negative societal impacts include potential malicious or unintended uses (e.g., disinformation, generating fake profiles, surveillance), fairness considerations (e.g., deployment of technologies that could make decisions that unfairly impact specific groups), privacy considerations, and security considerations.
- The conference expects that many papers will be foundational research and not tied to particular applications, let alone deployments. However, if there is a direct path to any negative applications, the authors should point it out. For example, it is legitimate to point out that an improvement in the quality of generative models could be used to generate deepfakes for disinformation. On the other hand, it is not needed to point out that a generic algorithm for optimizing neural networks could enable people to train models that generate Deepfakes faster.
- The authors should consider possible harms that could arise when the technology is being used as intended and functioning correctly, harms that could arise when the technology is being used as intended but gives incorrect results, and harms following from (intentional or unintentional) misuse of the technology.
- If there are negative societal impacts, the authors could also discuss possible mitigation strategies (e.g., gated release of models, providing defenses in addition to attacks, mechanisms for monitoring misuse, mechanisms to monitor how a system learns from feedback over time, improving the efficiency and accessibility of ML).

11. **Safeguards**

Question: Does the paper describe safeguards that have been put in place for responsible release of data or models that have a high risk for misuse (e.g., pretrained language models, image generators, or scraped datasets)?

Answer: [No]

Justification: The work described in the paper does not involve high-risk models or data that could pose significant misuse risks. Our gaze prediction model, designed for medical

image analysis, does not fall under categories such as pretrained language models or image generators with high potential for harmful or dual-use applications. However, we acknowledge the importance of responsible research practices.

While our model itself does not pose a direct misuse risk, we have taken care to ensure that all datasets used are either publicly available and ethically sourced or have been anonymized to prevent the disclosure of sensitive information. In cases where the technology could potentially be misapplied, such as in other domains outside medical image analysis, we encourage the community to implement safeguards for their respective models.

Guidelines:

- The answer NA means that the paper poses no such risks.
- Released models that have a high risk for misuse or dual-use should be released with necessary safeguards to allow for controlled use of the model, for example by requiring that users adhere to usage guidelines or restrictions to access the model or implementing safety filters.
- Datasets that have been scraped from the Internet could pose safety risks. The authors should describe how they avoided releasing unsafe images.
- We recognize that providing effective safeguards is challenging, and many papers do not require this, but we encourage authors to take this into account and make a best faith effort.

12. **Licenses for existing assets**

    Question: Are the creators or original owners of assets (e.g., code, data, models), used in the paper, properly credited and are the license and terms of use explicitly mentioned and properly respected?

    Answer: [Yes]

    Justification: All external assets used in this work, including datasets and models, are properly credited with clear citations and licensing terms.

    Guidelines:

    - The answer NA means that the paper does not use existing assets.
    - The authors should cite the original paper that produced the code package or dataset.
    - The authors should state which version of the asset is used and, if possible, include a URL.
    - The name of the license (e.g., CC-BY 4.0) should be included for each asset.
    - For scraped data from a particular source (e.g., website), the copyright and terms of service of that source should be provided.
    - If assets are released, the license, copyright information, and terms of use in the package should be provided. For popular datasets, `paperswithcode.com/datasets` has curated licenses for some datasets. Their licensing guide can help determine the license of a dataset.
    - For existing datasets that are re-packaged, both the original license and the license of the derived asset (if it has changed) should be provided.
    - If this information is not available online, the authors are encouraged to reach out to the asset's creators.

13. **New assets**

    Question: Are new assets introduced in the paper well documented and is the documentation provided alongside the assets?

    Answer: [Yes]

    Justification: Any new assets introduced in this paper, including custom datasets, models, or code, are well documented, and the documentation is provided alongside the assets.

    Guidelines:

    - The answer NA means that the paper does not release new assets.
    - Researchers should communicate the details of the dataset/code/model as part of their submissions via structured templates. This includes details about training, license, limitations, etc.

- The paper should discuss whether and how consent was obtained from people whose asset is used.
- At submission time, remember to anonymize your assets (if applicable). You can either create an anonymized URL or include an anonymized zip file.

14. **Crowdsourcing and research with human subjects**

    Question: For crowdsourcing experiments and research with human subjects, does the paper include the full text of instructions given to participants and screenshots, if applicable, as well as details about compensation (if any)?

    Answer: [NA]

    Justification: The paper does not involve crowdsourcing or research with human subjects. No such data collection or experimentation was conducted for the work presented.

    Guidelines:

    - The answer NA means that the paper does not involve crowdsourcing nor research with human subjects.
    - Including this information in the supplemental material is fine, but if the main contribution of the paper involves human subjects, then as much detail as possible should be included in the main paper.
    - According to the NeurIPS Code of Ethics, workers involved in data collection, curation, or other labor should be paid at least the minimum wage in the country of the data collector.

15. **Institutional review board (IRB) approvals or equivalent for research with human subjects**

    Question: Does the paper describe potential risks incurred by study participants, whether such risks were disclosed to the subjects, and whether Institutional Review Board (IRB) approvals (or an equivalent approval/review based on the requirements of your country or institution) were obtained?

    Answer: [NA]

    Justification: The paper does not involve research with human subjects, and thus no IRB approval or equivalent was required.

    Guidelines:

    - The answer NA means that the paper does not involve crowdsourcing nor research with human subjects.
    - Depending on the country in which research is conducted, IRB approval (or equivalent) may be required for any human subjects research. If you obtained IRB approval, you should clearly state this in the paper.
    - We recognize that the procedures for this may vary significantly between institutions and locations, and we expect authors to adhere to the NeurIPS Code of Ethics and the guidelines for their institution.
    - For initial submissions, do not include any information that would break anonymity (if applicable), such as the institution conducting the review.

16. **Declaration of LLM usage**

    Question: Does the paper describe the usage of LLMs if it is an important, original, or non-standard component of the core methods in this research? Note that if the LLM is used only for writing, editing, or formatting purposes and does not impact the core methodology, scientific rigorousness, or originality of the research, declaration is not required.

    Answer: [No]

    Justification: The core methodology and scientific rigor of the research do not involve LLMs as an important or original component. LLMs were not used in the development of the core methods.

    Guidelines:

    - The answer NA means that the core method development in this research does not involve LLMs as any important, original, or non-standard components.

- Please refer to our LLM policy (`https://neurips.cc/Conferences/2025/LLM`) for what should or should not be described.

