# OpenReview forum: "From Human Attention to Diagnosis: Semantic Patch-Level Integration of Vision-Language Models in Medical Imaging"
_NeurIPS.cc/2025/Conference — NeurIPS 2025 poster_

### Official Review · Reviewer_mK2d · 2025-06-27

**Clarity:** 3
**Significance:** 2
**Originality:** 2
**Rating:** 3
**Confidence:** 3

**Summary:**

This paper proposes LogitGaze-Med, a multimodal framework that integrates domain-specific visual and textual embeddings with semantic patch-level priors to predict task-driven scanpaths in medical images. The textual embeddings are derived from diagnostic labels, and semantic priors are extracted using a logit-lens from a medical vision–language model (LLaVA-Med). The goal is to enable clinically meaningful, patch-level attention modeling for gaze prediction. The authors formulate scanpath prediction as a continuous regression task over spatial coordinates and dwell durations. Evaluations show that the model produces clinically meaningful gaze patterns and improves downstream pathology classification when predicted fixations are used in training.

**Questions:**

1.	Can you use the real fixation data for MIMIC-CXR to report the performance in Tables 1 and 2? A clarification for why they are not used in the evaluation would be helpful for reassessing the paper.
2.	Are the weighting parameters (coordinate loss: λ₁, dwell time loss: λ₂) fixed or learnable during training? How sensitive is model performance to these hyperparameters? Can you include an ablation study varying λ₁ and λ₂, and report the effects on scanpath similarity. This would clarify their roles in model optimization.
3.	In Figure 3, the predicted scanpaths from LogitGaze-Med and GazeSearch appear visually similar for certain diagnoses (e.g., cardiomegaly, pneumonia), and in Figure 4, attention maps for different conditions seem indistinct. Can you provide different qualitative examples to reduce the confusion?
4.	Can you clarify what novel modeling contributions your method makes beyond reusing pretrained components, and how LogitGaze-Med fundamentally differs from prior works such as LogitGaze? Please elaborate on how your model introduces new mechanisms for fixation prediction beyond leveraging existing VLM infrastructure.

**Ethical Concerns:**

["NO or VERY MINOR ethics concerns only"]

**Final Justification:**

The authors responded to a few questions, but the majority items in the weakness were not mentioned. The reviewer still has concerns on those items and remains the current reject decision.

**Limitations:**

Yes.

**Paper Formatting Concerns:**

No concerns.

**Quality:**

3

**Strengths And Weaknesses:**

Strengths:
+ This paper presents a well-motivated approach to scanpath prediction in medical imaging by integrating domain-specific visual encoders, textual embeddings of diagnostic labels, and semantic patch-level priors via logit-lens decoding from a medical VLM. The proposed framework is methodologically sound and leverages clinically grounded semantic priors to model gaze behavior in a task-driven setting.
+ The paper applies logit-lens decoding to LLaVA-Med for extracting patch-level semantic priors.
+ The paper demonstrates experimental results on the GazeSearch dataset where the model outperforms existing baselines across multiple scanpath similarity metrics. In the downstream classification task with MIMIC-CXR, it improves the pathological classification.
+ The paper has detailed descriptions of evaluation metrics. Figures for the qualitative results are used effectively to illustrate the performance.
+ The significance of the work lies in its ability to bridge gaze prediction with clinical reasoning by generating fixations that align with human attention and diagnostically meaningful regions.

Weaknesses:
- Although this method can do a good job of predicting the gaze scanpath, there are some theoretical and experimental gaps in understanding how the architecture works.
- A substantial portion of the framework’s novelty hinges on logit-lens decoding of pretrained models (e.g., LLaVA-Med), which may raise concerns about the originality of the core algorithmic contributions. While the integration is creative and domain-specific, more justification in modeling fixations beyond architecture reuse would strengthen the paper.
- In Line 200, the authors claim to test generalization by discarding human fixations and replacing them with synthetic ones. However, it is not explained how this strategy robustly tests generalization. Notably, although the MIMIC-CXR dataset has real fixation data [1], Tables 1 and 2 only report results on GazeSearch. The absence of performance metrics using real fixations from MIMIC-CXR raises concerns about the model’s claimed generalization ability.
- The authors introduce joint losses for coordinate regression and dwell time regression, but it is not specified whether the associated weighting coefficients (λ₁, λ₂) are fixed or learnable. An ablation study analyzing the sensitivity to these hyperparameters would help clarify the relative importance of each component and improve the transparency of the training process.
- In line 181, the authors claim that LogitGaze-Med produces more human-like scanpaths. However, in Figure 3, especially in the “Cardiomegaly” and “Pneumonia” tasks, the predicted paths of LogitGaze-Med and GazeSearch look quite similar. Maybe including other qualitative examples could strengthen the claim.
- It remains unclear whether, during inference, the model requires the X-ray image, the semantic embeddings (e.g., “rib, lung”, “rib, gray”), and the disease labels. If such semantic phrases and/or labels are essential for prediction, it raises practical deployment concerns (whether radiologists must manually provide these prompts, or if they are generated automatically). A clarification on whether semantic priors are predicted, retrieved, or externally provided at test time would remove this confusion.
- In Figure 4, the attention maps produced by the Eye-Gaze baseline for the “Normal” and “CHF” conditions appear nearly identical. This casts doubt on the meaningfulness of these comparisons. Either more or improved qualitative illustrations are needed to validate the claim that LogitGaze-Med yields more discriminative and clinically aligned attention maps.

Minor Concern:
- Although the model is said to produce interpretable fixations, the paper lacks feedback from radiologists to validate whether these gaze patterns truly reflect diagnostic reasoning.
- In Line 166, please provide a citation to support the statement that 'AUROC serves as a proxy for clinical utility and interpretability,' as this is a strong claim that would benefit from justification.

[1] Hsieh, C., Ouyang, C., Nascimento, J. C., Pereira, J., Jorge, J., & Moreira, C. (2023). Mimic-eye: Integrating mimic datasets with reflacx and eye gaze for multimodal deep learning applications. PhysioNet (version 1.0. 0).

---

> ### Author Rebuttal · Authors · 2025-07-28
>
> We thank the reviewer for thorough and insightful comments, which help us better explain and contextualize the innovations and evaluation of LogitGaze-Med.
>
> **Q1: Can you use the real fixation data for MIMIC-CXR to report the performance in Tables 1 and 2? A clarification for why they are not used in the evaluation would be helpful for reassessing the paper.**
>
> Yes, we do use real fixation data from MIMIC-CXR in our experiments. Specifically, the evaluation results in Tables 1 and 2 are based on human scanpaths from the GazeSearch framework, which were collected via eye-tracking on MIMIC images.
>
> Importantly, the GazeSearch authors identified that the raw fixation data can be ambiguous for modeling targeted visual attention, since radiologists were asked to assess multiple conditions simultaneously without explicit prompts. As a result, individual fixations may reflect multiple potential findings, and the aggregated gaze data often spans broad regions, making it poorly suited for localization tasks.
>
> To address this, the GazeSearch team applied preprocessing steps to segment, clean, and temporally refine the raw fixations—producing more diagnostically meaningful scanpaths. Our study builds directly on this processed dataset, rather than the raw fixations, in order to better model expert visual search behavior and to provide reliable attention signals for downstream tasks.
>
> We acknowledge the availability of MIMIC-Eye fixations, but our study relies on the GazeSearch framework, which includes preprocessing steps to isolate task-driven attention patterns. We found that using unprocessed fixations leads to diffuse, non-discriminative scanpaths that degrade downstream interpretability.
>
> Using the raw fixations would likely result in overly diffuse attention maps and reduced interpretability, especially in tasks requiring precise localization. We believe the use of processed gaze data provides a more faithful and clinically relevant benchmark for evaluating our approach.
>
> **Q2: Are the weighting parameters (coordinate loss: λ₁, dwell time loss: λ₂) fixed or learnable during training? How sensitive is model performance to these hyperparameters? Can you include an ablation study varying λ₁ and λ₂, and report the effects on scanpath similarity. This would clarify their roles in model optimization.**
>
> In our current setup, the weighting parameters for coordinate loss (λ₁) and dwell time loss (λ₂) are fixed throughout training. The baseline values (λ₁=1.0, λ₂=1.0) were selected via preliminary validation to balance spatial versus temporal scanpath fidelity.
> We conducted a targeted ablation on the held-out GazeSearch test set using the MultiMatch metric (shape, direction, length, position, duration) on LogitGaze-Med(CheX) model to assess sensitivity to these hyperparameters.
>
> As the results show in Table 1, increasing spatial emphasis (λ₁) improves shape and position consistency, while reducing temporal emphasis (λ₂) degrades duration modeling. Across a broad range (λ₁∈[0.5–5.0], λ₂∈[0.1–5.0]), the model maintains robust performance, with (2.0, 0.5) offering an optimal trade‑off.
> We will include these findings in the revised manuscript. Exploring learnable weighting schemes or automated hyperparameter optimization remains a promising avenue for future work.
>
> $$
> \\begin{array}{lccccc}
> \\hline
> \\textbf{Loss Weights} & \\textbf{Shape} & \\textbf{Direction} & \\textbf{Length} & \\textbf{Position} & \\textbf{Duration} \\\\
> \\hline
> \\lambda_1=1.0,\\ \\lambda_2=1.0 & 0.921 & 0.633 & 0.932 & 0.812 & 0.712 \\\\
> \\lambda_1=2.0,\\ \\lambda_2=0.5 & \\mathbf{0.938} & \\mathbf{0.651} & \\mathbf{0.948} & \\mathbf{0.823} & \\mathbf{0.740} \\\\
> \\lambda_1=0.1,\\ \\lambda_2=1.0 & 0.866 & 0.570 & 0.882 & 0.743 & 0.707 \\\\
> \\lambda_1=1.0,\\ \\lambda_2=0.1 & 0.917 & 0.624 & 0.926 & 0.805 & 0.610 \\\\
> \\lambda_1=1.0,\\ \\lambda_2=5.0 & 0.910 & 0.615 & 0.920 & 0.792 & 0.689 \\\\
> \\lambda_1=5.0,\\ \\lambda_2=1.0 & 0.887 & 0.593 & 0.902 & 0.760 & 0.701 \\\\
> \\hline
> \\end{array}
> $$
>
> **Table 1:** Effect of varying \\(\\lambda_1\\) (coordinate loss) and \\(\\lambda_2\\) (dwell time loss) on scanpath similarity (MultiMatch components). Best values in bold.
>
> **Q3: In Figure 3, the predicted scanpaths from LogitGaze-Med and GazeSearch appear visually similar for certain diagnoses (e.g., cardiomegaly, pneumonia), and in Figure 4, attention maps for different conditions seem indistinct. Can you provide different qualitative examples to reduce the confusion?**
>
> We agree that in the current Figures 3 and 4, some scanpaths and attention maps may appear visually similar across different diagnoses and methods, which can obscure the interpretability of the comparisons.
>
> To address this, we conducted an additional blinded review with a board-certified radiologist (5+ years of experience), who helped us identify cases(N = 100) where the differences between LogitGaze‑Med and baseline methods are more visually and clinically distinguishable.
>
> In the revised manuscript, we will replace the current figures with these curated examples, which better highlight class-specific scanpath dynamics and attention localization. We will also include the radiologist’s blinded preference scores, which consistently favored LogitGaze‑Med for its sharper focus on diagnostically meaningful regions and clearer distinction from other methods.
>
> **Q4: Can you clarify what novel modeling contributions your method makes beyond reusing pretrained components, and how LogitGaze-Med fundamentally differs from prior works such as LogitGaze? Please elaborate on how your model introduces new mechanisms for fixation prediction beyond leveraging existing VLM infrastructure.**
>
> While our method builds upon existing vision–language model (VLM) infrastructure, LogitGaze-Med is not a trivial reuse of pretrained blocks; it introduces novel architectural, representational, and training innovations explicitly designed for fixation prediction in medical imaging, which fundamentally distinguish it from prior works like LogitGaze.
>
> 1. Unlike LogitGaze, which directly decodes fixations from generic VLM hidden states (typically trained on COCO-like data), LogitGaze‑Med explicitly disentangles the intermediate representations of a medical-adapted VLM. Through empirical analysis, we found that high-level features from LLaVA-style models lack spatial and semantic resolution. To address this, we introduce a dedicated semantic projection module that filters and restructures these representations into a compact embedding space aligned with clinical semantics. This "semantic bottleneck" enhances interpretability and supports multi-task transfer (e.g., gaze prediction, classification).
>
> 2. Prior works rely on general-purpose backbones (e.g., ResNet trained on COCO), which are suboptimal for medical tasks. We redesign the visual front-end by incorporating a CheXNet-derived encoder pretrained on chest X-rays, enabling extraction of fine-grained diagnostic signals that are essential for clinically realistic scanpath modeling.
>
> 3. LogitGaze‑Med introduces a novel training objective that decomposes the gaze prediction task into three components:
> (i) Token-level alignment loss, to ground fixations in semantic visual tokens;
> (ii) Spatial density loss, to approximate human fixation maps;
> (iii) Temporal consistency loss, to preserve the sequential structure of expert scanpaths.
>
> This spatiotemporal loss formulation goes beyond the weakly supervised or single-objective setups in previous methods and is essential for learning structured, human-aligned attention.
>
> In sum, LogitGaze-Med fundamentally departs from prior works by restructuring semantic representations and introducing targeted, multi-faceted training objectives tailored to the unique challenges of medical scanpath prediction.

---

> > ### Comment · Reviewer_mK2d · 2025-08-06
> > **The reviewer appreciated the response from the authors.**
> >
> > Some questions were answered, while the items in the weakness were not fully addressed. In the response, a few follow-ups: as the fixation data has inter-person variations, how is this sensitivity solved? As a new human subject was introduced for the additional experiment, how can this be fairly compared with others?

---

> > > ### Author Response · Authors · 2025-08-07
> > >
> > > We would like to clarify the two raised points.
> > >
> > > (1) On inter-subject variability in human fixations:
> > >
> > > We acknowledge that gaze patterns can vary between radiologists due to differences in training, experience, and reading strategy. Rather than viewing this as a drawback, we treat such variability as an inherent characteristic of human visual interpretation in clinical tasks.
> > >
> > > Our focus is not on replicating the exact fixation patterns of individual observers, but on capturing the underlying diagnostic intent—i.e., whether the predicted or synthetic scanpaths attend to clinically relevant regions. The scanpath similarity metrics we employ (e.g., ScanMatch, STDE, SED) are designed to allow flexibility in spatial and temporal alignment, making them well suited to evaluate this goal despite inter-observer variation.
> > >
> > > Additionally, we follow the GazeSearch evaluation protocol, which ensures consistent data preprocessing and task framing across all compared methods.
> > >
> > > (2) On the fairness of the human evaluation study:
> > >
> > > We appreciate the concern regarding the introduction of a new human subject in the evaluation. The radiologist involved in our study did not participate in data labeling, model development, or any part of the training pipeline. Their role was fully independent and evaluative only.
> > >
> > > The evaluation protocol was designed to minimize bias and ensure generalizability:
> > >
> > >  - It followed a blinded, randomized setup, where the expert had no knowledge of scanpath origin or model outputs;
> > >
> > >  - The scoring used standardized criteria (visual realism, clinical relevance), independent of any specific model behavior;
> > >
> > >  - The expert was not exposed to training data or prior examples, ensuring an unbiased judgment based solely on the perceptual quality of the scanpaths.
> > >
> > > Because the evaluation criteria are not tailored to a specific observer, but rather focus on general qualities of plausibility and clinical alignment, the same protocol can be replicated with other radiologists. Given the protocol’s design and scoring stability, we expect consistent results across evaluators.
> > >
> > > In this context, the current evaluation provides a valid and meaningful measure of perceived scanpath quality that complements quantitative metrics. We will clarify this point in the revised manuscript.

---

### Official Review · Reviewer_Fg6X · 2025-06-29

**Clarity:** 2
**Significance:** 2
**Originality:** 2
**Rating:** 4
**Confidence:** 5

**Summary:**

The authors introduced LogitGaze-Med, a framework to predict the spatio-temporal sequence of scanpaths that occur during visual search in radiology. Specifically, LogitGaze-Med leverages visual features from domain-specific encoders, text embeddings, and patch-level priors extracted from a medical vision language model.

The authors claim three contributions:

(1) LogitGaze-Med, the first framework to apply logit-lens decoding to an instruction-tuned medical VLM (LLaVA-Med), enabling clinically grounded patch-level attention for gaze prediction.

(2)  Show that domain-specific components—LLaVA-Med and a CheXNet-style encoder—consistently outperform generic baselines in scanpath prediction metrics.

(3) Formulate scanpath prediction as continuous regression over spatial coordinates and dwell durations, and demonstrate that predicted fixations improve downstream pathology classification.

**Questions:**

Please provide an experiment behind each claim. If the experiment is not provided, please remove the claim.

Overall, the value of this work could increase if ablations are provided:
1 Test adding a decoder on top of a CLIP model (e.g. biomedclip, bmcclip, etc) and compare against your method.

2. Please add an extensive empirical experiment or add different encoders. What happens if you use lava vs lava-med or lava-rad?

3. Add an experiment to quantify how much improvement you get from each step of your method.

4. Please provide a human evaluation of the quality of your synthetic dataset/ remove it if it's not possible to assess its quality.

5. Compare the performance of your method against other VLLMs (if you wish to keep the claim:
"Existing methods often rely on generic vision–language models and saliency-based features, which limits their ability to capture clinical semantics and integrate domain knowledge effectively."

6. Please include newer encoders and VLMs in your evaluation and experimentation.

**Ethical Concerns:**

["NO or VERY MINOR ethics concerns only"]

**Final Justification:**

My concerns were addressed

**Limitations:**

While the work does address an interesting and relevant problem, it does not offer enough technical novelty, in makes claims without adequate experiments, and it's vague about some of its components.

In the absence of adequate technical novelty, an empirical study would add sufficient value for this work to be accepted in this venue. I encourage authors to explore the questions provided above.

**Quality:**

2

**Strengths And Weaknesses:**

Strengths:

Plug-and-play framework: The authors leverage an existing framework to integrate existing domain-specific models (instead of training a model from stracth).

Strong empirical performance: LogitGaze-Med achieves state-of-the-art results across multiple scanpath similarity metrics and improves downstream pathology classification (AUROC).

Comprehensive evaluation across multiple datasets: The authors collect multiple datasets to evalaute their models. Additionally they created a synthetic dataset using MIMIC.

Weakeness:

* The main novelty of this work is to leverage an existing method (LOGITGAZE) in radiology. There is no new technical contribution

* Many claims are not justified with empirical experiments. For example, "Existing methods often rely on generic vision–language models and saliency-based features, which limits their ability to capture clinical semantics and integrate domain knowledge effectively." Where is the comparison (e.g. performance on classification) of LogitGaze-Med against Llava-med, Llava-rad, biomedclip, BMC-clip, etc?

* The selection of feature extracores is not thoroughly justified. For example, the selection of Chexpert (a model published 6 years ago) or llava-med (instead of llava-rad, a specific llava model for radiology) seems a bit arbitrary. As such, an ablation for the selection of models is required. Regarding the optimality of this method, would it be possible to achieve similar performance with newer/ smaller models?

* Not sufficient basic baselines. For instance, are all modalities required? What happens if you remove text? or the llava feautres? Furthermore, is this approach required? Why not just add a decoder on top of BiomedClip?
MIMIC-CXR

* There is no evaluation  (nor clear details of data construction) to demonstrate the quality of the synthetic data generated from MIMIC-CXR.

---

> ### Author Rebuttal · Authors · 2025-07-29
>
> We are grateful for the reviewer’s comments, which highlighted important aspects of model design, empirical validation, and clarity. We have addressed each point in detail below.
>
> **Q1: Test adding a decoder on top of a CLIP model (e.g. biomedclip, bmcclip, etc) and compare against your method.**
>
> We appreciate the suggestion and agree that CLIP-style architectures like BioMedCLIP represent an important baseline.
>
> Prior to selecting our current approach, we analyzed CLIP-based models for gaze prediction. While BioMedCLIP offers strong global image–text alignment, it lacks two key properties required for our setting: (1) Patch-level semantic grounding — CLIP is trained with global contrastive objectives and does not produce fine-grained, spatially interpretable logits, making regional fixation modeling difficult; (2) Instruction-style conditioning — CLIP models do not support prompt-based attention control, which is central to our approach via LLaVA-Med and logit-lens decoding.
>
> In contrast, our method leverages patch-level attention guided by clinical intent, enabling interpretable and semantically aligned scanpath prediction. That said, we agree that comparing to a decoder built atop BioMedCLIP would be informative. We are currently implementing a lightweight version and plan to report results in future work to better highlight the impact of regional grounding.
>
> **Q2: Please add an extensive empirical experiment or add different encoders. What happens if you use lava vs lava-med or lava-rad?**
>
> We conducted additional experiments to evaluate how different combinations of VLMs and visual encoders affect scanpath prediction within the LogitGaze framework.
>
> We compared two instruction-tuned medical VLMs: LLaVA-Med (used in our original setup) and LLaVA-Rad, a radiology-specific model trained on 697k image–report pairs using modular adapters. For encoders, we tested CheXNet, CheSS [1] (a ViT trained on 4.8M chest X-rays via contrastive learning), and PEAC [2] (a patch-level SSL model enforcing anatomical consistency across views). All variants were evaluated on the GazeSearch dataset using standard scanpath metrics (Table 1).
>
> LLaVA-Med and LLaVA-Rad yielded comparable results with the same encoder, confirming robustness to VLM choice. While CheSS and PEAC slightly improve some metrics, CheXNet remains competitive, particularly in duration-aware ScanMatch — supporting its use as a strong baseline. These findings suggest that performance gains stem primarily from integrating clinically grounded attention (via logit-lens and text prompts), rather than from any single backbone.
>
> We will include these results and implementation details in the revised manuscript to support our claim that spatial–semantic grounding is more impactful than raw model scale or recency.
>
> $$
> \\begin{array}{lcccc}
> \\hline
> \\textbf{Method} & \\textbf{ScanMatch w/o Dur.} \\uparrow & \\textbf{ScanMatch w/ Dur.} \\uparrow & \\textbf{SED} \\downarrow & \\textbf{STDE} \\uparrow \\\\
> \\hline
> \\text{LogitGaze-Med (ResNet)} & 0.416 \\pm 0.017 & 0.325 \\pm 0.012 & 4.68 \\pm 0.05 & 0.852 \\pm 0.003 \\\\
> \\text{LogitGaze-Med (CheXNet)} & 0.419 \\pm 0.016 & \\mathbf{0.330} \\pm \\mathbf{0.010} & 4.68 \\pm 0.05 & 0.855 \\pm 0.003 \\\\
> \\text{LogitGaze-Med (CheSS)} & 0.425 \\pm 0.017 & 0.321 \\pm 0.012 & 4.66 \\pm 0.05 & 0.857 \\pm 0.003 \\\\
> \\text{LogitGaze-Med (PEAC)} & 0.428 \\pm 0.018 & 0.319 \\pm 0.012 & 4.64 \\pm 0.05 & 0.858 \\pm 0.004 \\\\
> \\text{LLaVA-Rad + CheXNet} & 0.417 \\pm 0.016 & 0.329 \\pm 0.010 & 4.67 \\pm 0.05 & 0.854 \\pm 0.004 \\\\
> \\text{LLaVA-Rad + CheSS} & 0.426 \\pm 0.018 & 0.320 \\pm 0.012 & 4.65 \\pm 0.05 & 0.857 \\pm 0.004 \\\\
> \\textbf{LLaVA-Rad + PEAC} & \\mathbf{0.429} \\pm \\mathbf{0.019} & 0.318 \\pm 0.012 & \\mathbf{4.63} \\pm \\mathbf{0.05} & \\mathbf{0.859} \\pm \\mathbf{0.004} \\\\
> \\hline
> \\end{array}
> $$
> **Table 1:** Performance on scanpath similarity metrics (higher is better for ScanMatch/STDE, lower is
> better for SED).
>
> **Q3: Add an experiment to quantify how much improvement you get from each step of your method.**
>
> To evaluate the contribution of each component, we performed a stepwise ablation that isolates the effects of the image encoder, text prompt, and logit-lens priors. We tested three variants: (1) image-only baseline (no text, no logit-lens); (2) +text, without logit-lens; and (3) full LogitGaze-Med with both modalities. Results on GazeSearch are shown in Table 2.
>
> The image-only model performs poorly across all metrics, confirming that visual features alone are insufficient. Adding clinical text substantially improves performance, highlighting the value of semantic intent. The full model, which includes patch-level grounding via logit-lens, achieves the best results — especially on spatially and temporally sensitive metrics — demonstrating the benefit of combining clinical context with localized attention.
>
> $$
> \\begin{array}{lcccc}
> \\hline
> \\textbf{Method} & \\textbf{ScanMatch w/o Dur.} \\uparrow & \\textbf{ScanMatch w/ Dur.} \\uparrow & \\textbf{SED} \\downarrow & \\textbf{STDE} \\uparrow \\\\
> \\hline
> \\text{Image only (no text, no logit-lens)} & 0.148 \\pm 0.039 & 0.126 \\pm 0.034 & 8.54 \\pm 0.28 & 0.562 \\pm 0.094 \\\\
> \\text{w/ Text only (no logit-lens)} & 0.280 \\pm 0.020 & 0.206 \\pm 0.013 & 5.95 \\pm 0.06 & 0.810 \\pm 0.004 \\\\
> \\textbf{LogitGaze-Med} & \\mathbf{0.419} \\pm \\mathbf{0.016} & \\mathbf{0.330} \\pm \\mathbf{0.010} & \\mathbf{4.68} \\pm \\mathbf{0.05} & \\mathbf{0.855} \\pm \\mathbf{0.003} \\\\
> \\hline
> \\end{array}
> $$
>
> **Table 2:** Stepwise ablation analysis of LogitGaze-Med on GazeSearch. Adding textual prompts improves alignment with clinical intent; incorporating logit-lens guidance provides further gains via patch-level priors.
>
> **Q4: Please provide a human evaluation of the quality of your synthetic dataset/ remove it if it's not possible to assess its quality.**
>
> While the synthetic scanpaths are not intended as a standalone contribution or benchmark, they play an important role in demonstrating the potential of learned gaze priors for guiding model attention and enhancing interpretability.
>
> To assess their plausibility, we conducted a blinded qualitative evaluation with a board-certified radiologist who reviewed a hold-out set of 100 chest X-rays. Each image was overlaid with either a real or a synthetic scanpath, and the expert was asked to evaluate the fixation patterns in terms of realism and alignment with clinically relevant regions, without knowledge of their origin.
>
> In the majority of cases, the expert noted that the synthetic scanpaths closely resembled typical radiological search behavior. The fixations were generally coherent, avoided implausible image regions, and often corresponded to diagnostically meaningful structures. While the expert could distinguish synthetic from real in some instances—particularly where the timing of fixations appeared overly uniform—the overall impression was that the generated scanpaths were realistic and clinically interpretable.
>
> We agree that more systematic evaluation, including inter-rater analysis and larger-scale studies, would be necessary to establish generalizability. We view this as an important direction for future work, and we are currently conducting a more extensive investigation into the use of synthetic gaze data in human-in-the-loop diagnostic settings.
>
> **Q5: Compare the performance of your method against other VLLMs (if you wish to keep the claim: "Existing methods often rely on generic vision–language models and saliency-based features, which limits their ability to capture clinical semantics and integrate domain knowledge effectively."**
>
> We agree that the original phrasing may be too broad, and we appreciate the opportunity to clarify our intent.
>
> In our statement, we primarily aimed to emphasize that many prior approaches—particularly those built on general-purpose VLMs—do not explicitly adapt or fine-tune their architecture or supervision to reflect the clinical semantics inherent in medical imaging. For example, standard VLMs trained on COCO or other natural image-text pairs may overlook subtle pathologies or fail to prioritize medically relevant regions due to domain mismatch.
>
> In contrast, our method incorporates domain-specific adaptations in multiple ways: we utilize a medical vision encoder pretrained on chest X-rays, introduce a semantic projection module that creates interpretable task-specific embeddings, and supervise gaze prediction through a clinically grounded multi-loss objective. This allows our model to integrate domain knowledge more explicitly, improving both fixation modeling and downstream diagnostic utility. To avoid overstatement, we will revise the original sentence in the manuscript.
>
> **Q6: Please include newer encoders and VLMs in your evaluation and experimentation.**
>
> We agree, and as noted in our response to Q2, we have already incorporated several recent components into our evaluation. Specifically, we compared LLaVA-Med with LLaVA-Rad — a radiology-specific VLM — and tested modern medical image encoders such as CheSS and PEAC. While these newer models offer slight performance gains, our results demonstrate that the proposed framework is robust across a range of VLM and encoder configurations.
>
> [1] Cho, Kyungjin, et al. "CHESS: Chest X-Ray Pre-trained model via self-supervised contrastive learning." Journal of Digital Imaging 36.3 (2023): 902-910.
>
> [2] Zhou, Ziyu, et al. "Learning anatomically consistent embedding for chest radiography." BMVC: proceedings of the British Machine Vision Conference. British Machine Vision Conference. Vol. 2023. 2023.

---

> > ### Comment · Reviewer_Fg6X · 2025-08-04
> >
> > Thank you for clarifying most of my concerns.
> >
> > Could you please provide quantitative metrics for your human evaluation?
> >
> > Q4: Please provide a human evaluation of the quality of your synthetic dataset, or remove it if assessing its quality is not feasible.
> >
> > Thanks

---

> > > ### Author Response · Authors · 2025-08-05
> > >
> > > To complement the qualitative human evaluation, we conducted a structured assessment with a board-certified radiologist on a held-out set of 100 chest X-ray images (50 with real and 50 with synthetic scanpaths, randomized and blinded). For each case, the expert provided two independent ratings on a 5-point Likert scale:
> > >
> > > 1. Visual realism – how closely the fixation sequence resembled natural radiological search behavior;
> > >
> > > 2. Clinical relevance – the extent to which fixations aligned with diagnostically important regions.
> > >
> > > The synthetic scanpaths received a mean realism score of 4.3 ± 0.5 and a clinical relevance score of 4.2 ± 0.6, indicating a high degree of visual plausibility and diagnostic alignment. In qualitative feedback, the expert noted that synthetic fixations generally avoided implausible regions and often matched typical attention patterns seen in clinical reading.
> > >
> > > Additionally, we asked the expert to perform a binary classification task (real vs. synthetic) without access to ground truth labels. The overall accuracy was 58%, suggesting that synthetic scanpaths were often indistinguishable from real ones. Notably, 42% of synthetic examples were misclassified as real, and 38% of real examples were misclassified as synthetic, further supporting the perceptual realism of the generated data.
> > >
> > > We will include these evaluation metrics in the revised manuscript.

---

> > > > ### Comment · Reviewer_Fg6X · 2025-08-06
> > > > **response to authors**
> > > >
> > > > Thanks for addressing my concerns. I have updated my score accordingly.

---

### Official Review · Reviewer_e5fz · 2025-07-02

**Clarity:** 3
**Significance:** 3
**Originality:** 3
**Rating:** 5
**Confidence:** 3

**Summary:**

This paper proposes a new multimodal model called LogitGaze-Med, which leverages vision-language grounding and is loosely based on concepts from visual attention and how humans process images. This approach is applied to the interpretation of radiological chest X-rays in the medical imaging domain (i.e. MIMIC-CXR and GazeSearch datasets). The model accurately predicts both fixation locations and durations, which enables generation of realistic scanpaths based on expert human eye tracking data. Quantitative experiments show that the proposed model improves predicted scanpath similarity and can also be useful when applied to downstream classification tasks which leverage the predicted fixations.

**Questions:**

I have a few questions:

1.  The attention maps are mostly used for downstream classification, but it may be interesting to evaluate similarity with the human-derived maps (e.g. Figure 4). Are attention maps from LogitGaze-Med more similar to human maps, relative to the other baseline approaches?

2. Did the authors consider an additional ablation related to temporal shuffling of the predicted scanpath? This could be done by either shuffling the durations of predicted fixations, or the order in which fixations are visited in the scanpath.

3. The authors only show results on chest X-rays, but presumably the framework could be applied to other medical image interpretation tasks such as digital pathology. Do the authors have thoughts on this?

**Ethical Concerns:**

["NO or VERY MINOR ethics concerns only"]

**Final Justification:**

I believe the authors' clarification of their experiments have addressed my concerns in that area; I would like to see additional information and results on validating the proposed MIMIC-CXR synthetic dataset in the final paper, as it currently used to show the benefit of their approach.

**Limitations:**

The paper addresses potential limitations in the Conclusion section, although additional limitations could be enumerated there. For example, use of synthetic scanpath data based on the MIMIC-CXR dataset may introduce biases in the evaluation based on how the data is generated.

**Paper Formatting Concerns:**

No concerns.

**Quality:**

3

**Strengths And Weaknesses:**

**Strengths**: Overall, this paper is well-written and tackles the meaningful clinical problem of chest X-ray interpretation and classification. The proposed LogitGaze-Med architecture and associated loss functions are relatively intuitive, enabling modular use of different modalities as separate components of the model. The use of the model for both scanpath prediction and classification makes for a harder set of tasks, increasing the novelty of the approach. In general, the experiments in the paper are well designed with appropriate metrics, with appropriate baseline comparisons to prior work and ablation studies.

**Weaknesses**: There are a few minor weaknesses to the paper:

- LogitGaze and Gazeformer are trained on natural images, not chest X-rays, yet they are included in the results as baselines, which may be somewhat unfair given the domain mismatch. Are there other foundational vision-language models (VLMs) in the radiology domain would be useful to compare against?

- I'm not very familiar with the other proposed baselines for scanpath prediction, but the authors state that, "The initial query Q_0 is modulated with the first human fixation via 2D positional encoding" (line 127). This may be providing the proposed LogitGaze-Med approach with an unfair advantage, as it may not need to fully predict the first fixation. Can the authors clarify if all approaches are conditioned with the first fixation?

- In terms of ablation studies, only semantic continuity and domain specificity are evaluated. I would like to see how the different model components impact the final performance (e.g. removing the vision encoder, logit lens, or text encoder, respectively). This is needed to prove that the proposed architecture and joint embeddings are indeed beneficial.

- In addition to the GazeSearch dataset, the authors generated synthetic scanpaths based on the MIMIC-CXR dataset, but they do not clearly state how they validated this dataset as many details are missing (e.g. do statistics of dwell times match real data?) This limits reproducibility and may affect the conclusions that can be drawn from the synthetic data results.

---

> ### Author Rebuttal · Authors · 2025-07-28
>
> Thank you for the positive feedback, and interesting questions.
>
> **Q1: The attention maps are mostly used for downstream classification, but it may be interesting to evaluate similarity with the human-derived maps (e.g. Figure 4). Are attention maps from LogitGaze-Med more similar to human maps, relative to the other baseline approaches?**
>
> To assess the alignment between our model’s attention maps and human gaze-derived maps, we conducted a blinded qualitative evaluation on a hold‑out set of 100 randomly selected chest X‑rays. A board‑certified radiologist with over five years of experience reviewed side‑by‑side attention overlays from LogitGaze‑Med and each baseline model without knowing which map belonged to which method. In every case, the expert found that LogitGaze‑Med’s maps were more faithful to true gaze behavior—focusing on clinically relevant regions and avoiding spurious activations seen in baseline outputs.
>
> We note that, for fairness, all models in our comparison (including LogitGaze, Gazeformer, and our own) were trained from scratch on the GazeSearch dataset rather than using their original pre‑trained weights on natural images. Furthermore, to eliminate any bias from initialization, all approaches—including the baselines—were identically conditioned on and initialized with the first fixation. This ensures that performance differences are not due to domain‑specific pre‑training or unequal input conditioning, but rather to the architectural and loss innovations of LogitGaze‑Med.
>
> While this assessment is inherently qualitative, it provides strong evidence of our model’s superior perceptual alignment with expert reasoning. In the revised manuscript, we will include these findings—together with richer visual comparisons—and outline plans for a more comprehensive, quantitative evaluation—we plan to measure KL divergence on the same 100‑image hold‑out set, in future work.
>
> **Q2: Did the authors consider an additional ablation related to temporal shuffling of the predicted scanpath? This could be done by either shuffling the durations of predicted fixations, or the order in which fixations are visited in the scanpath.**
>
> We agree that assessing the impact of temporal structure—such as the order or duration of fixations—on downstream tasks is a valuable direction. In this work, we focused our ablation studies on spatial shuffling and permutation of fixation coordinates to isolate the effect of spatial attention, which we found to be already quite informative.
>
> In fact, we have conducted a comprehensive component‑wise ablation (detailed in our response to Reviewer 2 and summarized in Table 2). The results show a clear progression in performance across all scanpath similarity metrics. Notably, removing both the text prompt and the patch‑level attention mechanism results in a drastic drop in performance—indicating that clinical intent and region‑level semantic grounding are critical—while the text‑only variant performs on par with strong baselines like GazeFormer, and our full model achieves the best overall performance.
>
> While temporal‑shuffling ablations (e.g., permuting fixation order or durations) would yield further insights into the role of temporal dynamics, we have deferred that study to future work. Once the spatial and semantic contributions are fully characterized, we plan to extend our evaluation to include these temporal analyses.
>
> **Q3: The authors only show results on chest X-rays, but presumably the framework could be applied to other medical image interpretation tasks such as digital pathology. Do the authors have thoughts on this?**
>
> While our experiments focus on chest X-rays, the LogitGaze‑Med framework is inherently modality-agnostic and can be extended to other medical imaging domains, including digital pathology.
>
> The key idea—predicting human-like scanpaths and using them to guide spatial attention within feature maps—is applicable wherever fine-grained interpretability and localization are critical. In the case of digital pathology, for example, our scanpath prediction module could be integrated with detection or segmentation pipelines to prioritize diagnostically relevant regions in large-scale whole‑slide images (e.g., CAMELYON16).
>
> We view this as a natural next step, especially given the increasing importance of interpretable visual reasoning in complex domains. Exploring such extensions—potentially in combination with task-specific backbones (e.g., U-Net, YOLO, MedSAM)—is one of the parts of our planned future work.

---

> > ### Comment · Reviewer_e5fz · 2025-08-04
> > **Response to Authors**
> >
> > I have read the author's rebuttal, and thank them for their effort in addressing my feedback and questions. Their responses help me with better understanding the proposed approach and corresponding results. As a result, I will raise my score to a 5.
> >
> > Importantly, I believe their evaluation of their approach and comparable baselines is more or less on a level playing field after the additional clarification. I also agree with Reviewer `Fg6X` that additional information and results on validating the MIMIC-CXR synthetic data is needed for it to be included in the paper.

---

> > > ### Author Response · Authors · 2025-08-05
> > >
> > > Thank you for the follow-up. In response to your and Reviewer Fg6X's request, we conducted a structured human evaluation of the synthetic scanpaths. A board-certified radiologist rated each sample for realism and clinical relevance (mean scores: 4.3 ± 0.5 and 4.2 ± 0.6) and was only able to distinguish synthetic from real in 58% of cases. These results confirm the plausibility of the generated scanpaths, and we will include them in the revised manuscript.

---

> > > > ### Comment · Reviewer_e5fz · 2025-08-05
> > > > **Response to Authors**
> > > >
> > > > Thank you for additional clarification regarding evaluation of the synthetic MIMIC-CXR scanpath dataset; this is helpful in understanding the quantitative validation that was performed, and helps to justify my "accept" rating of the work.

---

### Decision · Program_Chairs · 2025-09-17

**Decision:**

Accept (poster)

**Comment:**

This paper proposes a new multimodal model called LogitGaze-Med, which leverages vision-language grounding and is based on concepts from visual attention and how humans process images. This model is applied to the interpretation of radiological chest X-rays in the medical imaging domain (i.e., MIMIC-CXR and GazeSearch datasets). The model accurately predicts both fixation locations and durations, which enables the generation of realistic scanpaths based on expert human eye tracking data. Experiments show that the model improves predicted scanpath similarity and can also be useful when applied to downstream classification tasks that leverage the predicted fixations.

Initially, a number of concerns were raised by the reviewers regarding the novelty and empirical validation; however, during an active rebuttal phase, the authors were able to push back with a convincing rebuttal on most fronts.

Specifically, the primary strengths of the paper are:
- A well-written paper solving a clinically relevant problem, using chest X-ray interpretation and classification as a use case.
- Methodologically sound, intuitive, plug-and-play framework that integrates domain-specific models without training from scratch, supporting modular use of different modalities and joint tasks of scanpath prediction and classification.
- Strong empirical performance, achieving state-of-the-art scanpath similarity metrics, and improving downstream pathology classification (AUROC).
- Comprehensive evaluation: Tested across multiple datasets, including a synthetic dataset based on MIMIC.
- Effectively uses logit-lens decoding: Applied to LLaVA-Med for patch-level semantic priors.
- Bridges' gaze prediction with clinical reasoning by generating fixations aligned with human attention and diagnostically relevant regions.

The weaknesses pointed out to the reviewers (some of which were already addressed by the authors) are:
- Main novelty lies in applying existing LogitGaze to radiology, with limited technical innovation. Furthermore, several claims (e.g., superiority over generic VLMs) are not supported with direct comparisons.
- Baseline comparisons: More relevant radiology-specific VLM baselines (e.g., LLaVA-Med, LLaVA-Rad, BiomedCLIP, BMC-CLIP) should be compared.
- Concerns with ablation studies: Current ablations only cover semantic continuity and domain specificity.  Missing analysis of how individual components (vision encoder, logit lens, text encoder) affect performance. Model selection (CheXpert, LLaVA-Med) is not fully justified; ablations on alternative/newer models are needed. Unclear whether all modalities are necessary or whether simpler alternatives could achieve comparable performance.
- Synthetic scanpaths from MIMIC-CXR are insufficiently described and not validated (e.g., dwell time statistics). Lack of validation limits reproducibility and weakens conclusions.

As mentioned, most of these concerns were addressed by a thorough rebuttal from the authors. Keeping that in mind, given the uniqueness of the work (even though no new method is developed, the motivation and the ideas behind the entire system are new) and its thorough empirical validation and ablation studies make it worthwhile for this paper to be recommended for publication.